# MISSO: Minimization by Incremental Stochastic Surrogate Optimization for Large Scale Nonconvex and Nonsmooth Problems

## Abstract

Many constrained, nonconvex and nonsmooth optimization problems can be tackled using the majorization-minimization (MM) method which alternates between constructing a surrogate function which upper bounds the objective function, and then minimizing this surrogate. For problems which minimize a finite sum of functions, a stochastic version of the MM method selects a batch of functions at random at each iteration and optimizes the accumulated surrogate. However, in many cases of interest such as variational inference for latent variable models, the surrogate functions are expressed as an expectation. In this contribution, we propose a doubly stochastic MM method based on Monte Carlo approximation of these stochastic surrogates. We establish asymptotic and non-asymptotic convergence of our scheme in a constrained, nonconvex, nonsmooth optimization setting. We apply our new framework for inference of logistic regression model with missing data and for variational inference of Bayesian variants of LeNet-5 and Resnet-18 on respectively the MNIST and CIFAR-10 datasets.

## 1 Introduction

We consider the *constrained* minimization problem of a finite sum of functions:

$$\min_{\boldsymbol{\theta} \in \Theta} \ \mathcal{L}(\boldsymbol{\theta}) := \frac{1}{n} \sum_{i=1}^{n} \mathcal{L}_i(\boldsymbol{\theta}) \ , \tag{1}$$

where $\Theta$ is a convex, compact, and closed subset of $\mathbb{R}^p$, and for any $i \in [\![1, n]\!]$, the function $\mathcal{L}_i : \mathbb{R}^p \to \mathbb{R}$ is bounded from below and is (possibly) nonconvex and nonsmooth.

To tackle the optimization problem (1), a popular approach is to apply the majorization-minimization (MM) method which iteratively minimizes a majorizing surrogate function. A large number of existing procedures fall into this general framework, for instance gradient-based or proximal methods or the Expectation-Maximization (EM) algorithm (McLachlan & Krishnan, 2008) and some variational Bayes inference techniques (Jordan et al., 1999); see for example (Razaviyayn et al., 2013) and (Lange, 2016) and the references therein. When the number of terms $n$ in (1) is large, the vanilla MM method may be intractable because it requires to construct a surrogate function for all the $n$ terms $\mathcal{L}_i$ at each iteration. Here, a remedy is to apply the Minimization by Incremental Surrogate Optimization (MISO) method proposed by Mairal (2015), where the surrogate functions are updated incrementally. The MISO method can be interpreted as a combination of MM and ideas which have emerged for variance reduction in stochastic gradient methods (Schmidt et al., 2017). An extended analysis of MISO has been proposed in (Qian et al., 2019).

The success of the MISO method rests upon the efficient minimization of surrogates such as convex functions, see (Mairal, 2015, Section 2.3). A notable application of MISO-like algorithms is described in (Mensch et al., 2017) where the authors builds upon the stochastic majorization-minimization framework of Mairal (2015) to introduce a method for sparse matrix factorization. Yet, in many applications of interest, the natural surrogate functions are intractable, yet they are defined as expectation of tractable functions. For instance, this is the case for inference in latent variable models via maximum likelihood (McLachlan & Krishnan, 2008). Another application is

variational inference (Ghahramani, 2015), in which the goal is to approximate the posterior distribution of parameters given the observations; see for example (Neal, 2012; Blundell et al., 2015; Polson et al., 2017; Rezende et al., 2014; Li & Gal, 2017).

This paper fills the gap in the literature by proposing a method called *Minimization by Incremental Stochastic Surrogate Optimization (MISSO)*, designed for the nonconvex and nonsmooth finite sum optimization, with a finite-time convergence guarantee. Our work aims at formulating a *generic class* of incremental stochastic surrogate methods for nonconvex optimization and building the theory to understand its behavior. In particular, we provide convergence guarantees for stochastic EM and Variational Inference-type methods, under mild conditions. In summary, our contributions are:

- we propose a unifying framework of analysis for incremental stochastic surrogate optimization when the surrogates are defined as expectations of tractable functions. The proposed MISSO method is built on the Monte Carlo integration of the intractable surrogate function, *i.e.,* a doubly stochastic surrogate optimization scheme.

- we present an incremental update of the commonly used variational inference and Monte Carlo EM methods as special cases of our newly introduced framework. The analysis of those two algorithms is thus conducted under this unifying framework of analysis.

- we establish both asymptotic and non-asymptotic convergence for the MISSO method. In particular, the MISSO method converges almost surely to a stationary point and in $\mathcal{O}(n/\epsilon)$ iterations to an $\epsilon$-stationary point, see Theorem 1.

- in essence, we relax the class of surrogate functions used in MISO (Mairal, 2015) and allow for intractable surrogates that can only be evaluated by Monte-Carlo approximations. Working at the crossroads of *Optimization* and *Sampling* constitutes what we believe to be the novelty and the technicality of our framework and theoretical results.

In Section 2, we review the techniques for incremental minimization of finite sum functions based on the MM principle; specifically, we review the MISO method (Mairal, 2015), and present a class of surrogate functions expressed as an expectation over a latent space. The MISSO method is then introduced for the latter class of intractable surrogate functions requiring approximation. In Section 3, we provide the asymptotic and non-asymptotic convergence analysis for the MISSO method (and of the MISO (Mairal, 2015) one as a special case). Section 4 presents numerical applications including parameter inference for logistic regression with missing data and variational inference for two types of Bayesian neural networks. The proofs of theoretical results are reported as Supplement.

**Notations.** We denote $[\![1, n]\!] = \{1, \ldots, n\}$. Unless otherwise specified, $\| \cdot \|$ denotes the standard Euclidean norm and $\langle \cdot \, | \, \cdot \rangle$ is the inner product in the Euclidean space. For any function $f : \Theta \to \mathbb{R}$, $f'(\boldsymbol{\theta}, \boldsymbol{d})$ is the directional derivative of $f$ at $\boldsymbol{\theta}$ along the direction $\boldsymbol{d}$, *i.e.,*

$$f'(\boldsymbol{\theta}, \boldsymbol{d}) := \lim_{t \to 0^+} \frac{f(\boldsymbol{\theta} + t\boldsymbol{d}) - f(\boldsymbol{\theta})}{t} . \tag{2}$$

The directional derivative is assumed to exist for the functions introduced throughout this paper.

## 2 INCREMENTAL MINIMIZATION OF FINITE SUM NONCONVEX FUNCTIONS

The objective function in (1) is composed of a finite sum of possibly nonsmooth and nonconvex functions. A popular approach here is to apply the MM method, which tackles (1) through alternating between two steps — (i) minimizing a *surrogate* function which upper bounds the original objective function; and (ii) updating the surrogate function to tighten the upper bound.

As mentioned in the introduction, the MISO method (Mairal, 2015) is developed as an iterative scheme that only updates the surrogate functions *partially* at each iteration. Formally, for any $i \in [\![1, n]\!]$, we consider a surrogate function $\widehat{\mathcal{L}}_i(\boldsymbol{\theta}; \overline{\boldsymbol{\theta}})$ which satisfies the assumptions (**H1**, **H2**):

**H1.** *For all $i \in [\![1, n]\!]$ and $\overline{\boldsymbol{\theta}} \in \Theta$, $\widehat{\mathcal{L}}_i(\boldsymbol{\theta}; \overline{\boldsymbol{\theta}})$ is convex w.r.t. $\boldsymbol{\theta}$, and it holds*

$$\widehat{\mathcal{L}}_i(\boldsymbol{\theta}; \overline{\boldsymbol{\theta}}) \geq \mathcal{L}_i(\boldsymbol{\theta}), \ \forall \, \boldsymbol{\theta} \in \Theta , \tag{3}$$

*where the equality holds when $\boldsymbol{\theta} = \overline{\boldsymbol{\theta}}$.*

**H2.** *For any* $\overline{\boldsymbol{\theta}}_i \in \Theta$, $i \in [\![1, n]\!]$ *and some* $\epsilon > 0$, *the difference function* $\widehat{e}(\boldsymbol{\theta}; \{\overline{\boldsymbol{\theta}}_i\}_{i=1}^n) := \frac{1}{n} \sum_{i=1}^n \widehat{\mathcal{L}}_i(\boldsymbol{\theta}; \overline{\boldsymbol{\theta}}_i) - \mathcal{L}(\boldsymbol{\theta})$ *is defined for all* $\boldsymbol{\theta} \in \Theta_\epsilon$ *and differentiable for all* $\boldsymbol{\theta} \in \Theta$, *where* $\Theta_\epsilon = \{\boldsymbol{\theta} \in \mathbb{R}^d, \inf_{\boldsymbol{\theta}' \in \Theta} \|\boldsymbol{\theta} - \boldsymbol{\theta}'\| < \epsilon\}$ *is an* $\epsilon$-*neighborhood set of* $\Theta$. *Moreover, for some constant* $L$, *the gradient satisfies*

$$\|\nabla \widehat{e}(\boldsymbol{\theta}; \{\overline{\boldsymbol{\theta}}_i\}_{i=1}^n)\|^2 \leq 2L \widehat{e}(\boldsymbol{\theta}; \{\overline{\boldsymbol{\theta}}_i\}_{i=1}^n), \ \forall \, \boldsymbol{\theta} \in \Theta \ . \tag{4}$$

We remark that H1 is a common assumption used for surrogate functions, see (Mairal, 2015, Section 2.3). H2 can be satisfied when the difference function $\widehat{e}(\boldsymbol{\theta}; \{\overline{\boldsymbol{\theta}}_i\}_{i=1}^n)$ is $L$-smooth, *i.e.*, $\widehat{e}$ is differentiable on $\Theta$ and its gradient $\nabla \widehat{e}$ is L-Lipschitz, $\forall \boldsymbol{\theta} \in \Theta$. H2 can be implied by applying (Razaviyayn et al., 2013, Proposition 1).

The inequality (3) implies $\widehat{\mathcal{L}}_i(\boldsymbol{\theta}; \overline{\boldsymbol{\theta}}) \geq \mathcal{L}_i(\boldsymbol{\theta}) > -\infty$ for any $\boldsymbol{\theta} \in \Theta$. The MISO method is an incremental version of the MM method, as summarized by Algorithm 1, which shows that the MISO method maintains an iteratively updated set of upper-bounding surrogate functions $\{\mathcal{A}_i^k(\boldsymbol{\theta})\}_{i=1}^n$ and updates the iterate via minimizing the average of the surrogate functions.

---

**Algorithm 1** The MISO method (Mairal, 2015).

1: **Input:** initialization $\boldsymbol{\theta}^{(0)}$.
2: Initialize the surrogate function as
$\mathcal{A}_i^0(\boldsymbol{\theta}) := \widehat{\mathcal{L}}_i(\boldsymbol{\theta}; \boldsymbol{\theta}^{(0)})$, $i \in [\![1, n]\!]$.
3: **for** $k = 0, 1, ..., K_{\max}$ **do**
4:   Pick $i_k$ uniformly from $[\![1, n]\!]$.
5:   Update $\mathcal{A}_i^{k+1}(\boldsymbol{\theta})$ as:

$$\mathcal{A}_i^{k+1}(\boldsymbol{\theta}) = \begin{cases} \widehat{\mathcal{L}}_i(\boldsymbol{\theta}; \boldsymbol{\theta}^{(k)}), & \text{if } i = i_k \\ \mathcal{A}_i^k(\boldsymbol{\theta}), & \text{otherwise.} \end{cases}$$

6:   Set $\boldsymbol{\theta}^{(k+1)} \in \arg\min_{\boldsymbol{\theta} \in \Theta} \frac{1}{n} \sum_{i=1}^n \mathcal{A}_i^{k+1}(\boldsymbol{\theta})$.
7: **end for**

---

Particularly, only one out of the $n$ surrogate functions is updated at each iteration [cf. Line 5] and the sum function $\frac{1}{n} \sum_{i=1}^n \mathcal{A}_i^{k+1}(\boldsymbol{\theta})$ is designed to be 'easy to optimize', which, for example, can be a sum of quadratic functions. As such, the MISO method is suitable for large-scale optimization as the computation cost per iteration is independent of $n$. Under H1, H2, it was shown that the MISO method converges almost surely to a stationary point of (1) (Mairal, 2015, Prop. 3.1).

We now consider the case when the surrogate functions $\widehat{\mathcal{L}}_i(\boldsymbol{\theta}; \overline{\boldsymbol{\theta}})$ are intractable. Let $\mathsf{Z}$ be a measurable set, $p_i : \mathsf{Z} \times \Theta \to \mathbb{R}_+$ a probability density function, $r_i : \Theta \times \Theta \times \mathsf{Z} \to \mathbb{R}$ a measurable function and $\mu_i$ a $\sigma$-finite measure. We consider surrogate functions which satisfy H1, H2 and that can be expressed as an expectation, *i.e.*:

$$\widehat{\mathcal{L}}_i(\boldsymbol{\theta}; \overline{\boldsymbol{\theta}}) := \int_{\mathsf{Z}} r_i(\boldsymbol{\theta}; \overline{\boldsymbol{\theta}}, z_i) p_i(z_i; \overline{\boldsymbol{\theta}}) \mu_i(dz_i) \quad \forall (\boldsymbol{\theta}, \overline{\boldsymbol{\theta}}) \in \Theta \times \Theta \ . \tag{5}$$

Plugging (5) into the MISO method is not feasible since the update step in Step 6 involves a minimization of an expectation. Several motivating examples of (1) are given in Section 2.

In this paper, we propose the *Minimization by Incremental Stochastic Surrogate Optimization* (MISSO) method which replaces the expectation in (5) by *Monte Carlo* integration and then optimizes the objective function (1) in an incremental manner. Denote by $M \in \mathbb{N}$ the Monte Carlo batch size and let $\{z_m \in \mathsf{Z}\}_{m=1}^M$ be a set of samples. These samples can be drawn (Case 1) i.i.d. from the distribution $p_i(\cdot; \overline{\boldsymbol{\theta}})$ or (Case 2) from a Markov chain with stationary distribution $p_i(\cdot; \overline{\boldsymbol{\theta}})$; see Section 3 for illustrations. To this end, we define the stochastic surrogate as follows:

$$\widetilde{\mathcal{L}}_i(\boldsymbol{\theta}; \overline{\boldsymbol{\theta}}, \{z_m\}_{m=1}^M) := \frac{1}{M} \sum_{m=1}^M r_i(\boldsymbol{\theta}; \overline{\boldsymbol{\theta}}, z_m) \ , \tag{6}$$

and we summarize the proposed MISSO method in Algorithm 2. Compared to the MISO method, there is a crucial difference in that the MISSO method involves two types of randomness. The first level of randomness comes from the selection of $i_k$ in Line 5. The second level of randomness stems from the set of Monte Carlo approximated functions $\widetilde{\mathcal{A}}_i^k(\boldsymbol{\theta})$ used in lieu of $\mathcal{A}_i^k(\boldsymbol{\theta})$ in Line 6 when optimizing for the next iterate $\boldsymbol{\theta}^{(k)}$. We now discuss two applications of the MISSO method.

**Example 1: Maximum Likelihood Estimation for Latent Variable Model.** Latent variable models (Bishop, 2006) are constructed by introducing unobserved (latent) variables which help explain the observed data. We consider $n$ independent observations $((y_i, z_i), i \in [\![n]\!])$ where $y_i$ is observed and $z_i$ is latent. In this incomplete data framework, define $\{f_i(z_i, \boldsymbol{\theta}), \boldsymbol{\theta} \in \Theta\}$ to be the complete

---

**Algorithm 2** The MISSO method.

1: **Input:** initialization $\boldsymbol{\theta}^{(0)}$; a sequence of non-negative numbers $\{M_{(k)}\}_{k=0}^{\infty}$.
2: For all $i \in [\![1, n]\!]$, draw $M_{(0)}$ Monte Carlo samples with the stationary distribution $p_i(\cdot; \boldsymbol{\theta}^{(0)})$.
3: Initialize the surrogate function as

$$\widetilde{\mathcal{A}}_i^0(\boldsymbol{\theta}) := \widetilde{\mathcal{L}}_i(\boldsymbol{\theta}; \boldsymbol{\theta}^{(0)}, \{z_{i,m}^{(0)}\}_{m=1}^{M_{(0)}}), \ i \in [\![1, n]\!] \ .$$

4: **for** $k = 0, 1, ..., K_{\mathsf{max}}$ **do**
5:     Pick a function index $i_k$ uniformly on $[\![1, n]\!]$.
6:     Draw $M_{(k)}$ Monte Carlo samples with the stationary distribution $p_i(\cdot; \boldsymbol{\theta}^{(k)})$.
7:     Update the individual surrogate functions recursively as:

$$\widetilde{\mathcal{A}}_i^{k+1}(\boldsymbol{\theta}) = \begin{cases} \widetilde{\mathcal{L}}_i(\boldsymbol{\theta}; \boldsymbol{\theta}^{(k)}, \{z_{i,m}^{(k)}\}_{m=1}^{M_{(k)}}), & \text{if } i = i_k \\ \widetilde{\mathcal{A}}_i^k(\boldsymbol{\theta}), & \text{otherwise.} \end{cases}$$

8:     Set $\boldsymbol{\theta}^{(k+1)} \in \arg\min_{\boldsymbol{\theta} \in \Theta} \widetilde{\mathcal{L}}^{(k+1)}(\boldsymbol{\theta}) := \frac{1}{n} \sum_{i=1}^{n} \widetilde{\mathcal{A}}_i^{k+1}(\boldsymbol{\theta})$.
9: **end for**

---

data likelihood models, *i.e.,* the joint likelihood of the observations and latent variables. Let

$$g_i(\boldsymbol{\theta}) := \int_{\mathsf{Z}} f_i(z_i, \boldsymbol{\theta}) \mu_i(\mathrm{d}z_i), \ i \in [\![1, n]\!], \ \boldsymbol{\theta} \in \Theta$$

denote the incomplete data likelihood, *i.e.,* the marginal likelihood of the observations $y_i$. For ease of notations, the dependence on the observations is made implicit. The maximum likelihood (ML) estimation problem sets the individual objective function $\mathcal{L}_i(\boldsymbol{\theta})$ to be the $i$-th negated incomplete data log-likelihood $\mathcal{L}_i(\boldsymbol{\theta}) := -\log g_i(\boldsymbol{\theta})$.

Assume, without loss of generality, that $g_i(\boldsymbol{\theta}) \neq 0$ for all $\boldsymbol{\theta} \in \Theta$. We define by $p_i(z_i, \boldsymbol{\theta}) := f_i(z_i, \boldsymbol{\theta})/g_i(\boldsymbol{\theta})$ the conditional distribution of the latent variable $z_i$ given the observations $y_i$. A surrogate function $\widehat{\mathcal{L}}_i(\boldsymbol{\theta}; \overline{\boldsymbol{\theta}})$ satisfying H1 can be obtained through writing $f_i(z_i, \boldsymbol{\theta}) = \frac{f_i(z_i, \boldsymbol{\theta})}{p_i(z_i, \overline{\boldsymbol{\theta}})} p_i(z_i, \overline{\boldsymbol{\theta}})$ and applying the Jensen inequality:

$$\widehat{\mathcal{L}}_i(\boldsymbol{\theta}; \overline{\boldsymbol{\theta}}) = \int_{\mathsf{Z}} \underbrace{\log\left(p_i(z_i, \overline{\boldsymbol{\theta}})/f_i(z_i, \boldsymbol{\theta})\right)}_{= r_i(\boldsymbol{\theta}; \overline{\boldsymbol{\theta}}, z_i)} p_i(z_i, \overline{\boldsymbol{\theta}}) \mu_i(\mathrm{d}z_i) \ . \tag{7}$$

We note that H2 can also be verified for common distribution models. We can apply the MISSO method following the above specification of $r_i(\boldsymbol{\theta}; \overline{\boldsymbol{\theta}}, z_i)$ and $p_i(z_i, \overline{\boldsymbol{\theta}})$.

**Example 2: Variational Inference.** Let $((x_i, y_i), i \in [\![1, n]\!])$ be i.i.d. input-output pairs and $w \in \mathsf{W} \subseteq \mathbb{R}^d$ be a latent variable. When conditioned on the input data $x = (x_i, i \in [\![1, n]\!])$, the joint distribution of $y = (y_i, i \in [\![1, n]\!])$ and $w$ is given by:

$$p(y, w | x) = \pi(w) \prod_{i=1}^{n} p(y_i | x_i, w) \ . \tag{8}$$

Our goal is to compute the posterior distribution $p(w|y, x)$. In most cases, the posterior distribution $p(w|y, x)$ is intractable and is approximated using a family of parametric distributions, $\{q(w, \boldsymbol{\theta}), \boldsymbol{\theta} \in \Theta\}$. The variational inference (VI) problem (Blei et al., 2017) boils down to minimizing the Kullback-Leibler (KL) divergence between $q(w, \boldsymbol{\theta})$ and the posterior distribution $p(w|y, x)$:

$$\min_{\boldsymbol{\theta} \in \Theta} \mathcal{L}(\boldsymbol{\theta}) := \mathrm{KL}\left(q(w; \boldsymbol{\theta}) || p(w|y, x)\right) := \mathbb{E}_{q(w; \boldsymbol{\theta})}\left[\log\left(q(w; \boldsymbol{\theta})/p(w|y, x)\right)\right] \ . \tag{9}$$

Using (8), we decompose $\mathcal{L}(\boldsymbol{\theta}) = n^{-1} \sum_{i=1}^{n} \mathcal{L}_i(\boldsymbol{\theta}) + \text{const.}$ where:

$$\mathcal{L}_i(\boldsymbol{\theta}) := -\mathbb{E}_{q(w; \boldsymbol{\theta})}\left[\log p(y_i | x_i, w)\right] + \frac{1}{n} \mathbb{E}_{q(w; \boldsymbol{\theta})}\left[\log q(w; \boldsymbol{\theta})/\pi(w)\right] := r_i(\boldsymbol{\theta}) + d(\boldsymbol{\theta}) \ . \tag{10}$$

Directly optimizing the finite sum objective function in (9) can be difficult. First, with $n \gg 1$, evaluating the objective function $\mathcal{L}(\boldsymbol{\theta})$ requires a full pass over the entire dataset. Second, for some

complex models, the expectations in (10) can be intractable even if we assume a simple parametric model for $q(w; \boldsymbol{\theta})$. Assume that $\mathcal{L}_i$ is L-smooth. We apply the MISSO method with a quadratic surrogate function defined as:

$$\widehat{\mathcal{L}}_i(\boldsymbol{\theta}; \overline{\boldsymbol{\theta}}) := \mathcal{L}_i(\overline{\boldsymbol{\theta}}) + \left\langle \nabla_{\boldsymbol{\theta}} \mathcal{L}_i(\overline{\boldsymbol{\theta}}) \, | \, \boldsymbol{\theta} - \overline{\boldsymbol{\theta}} \right\rangle + \frac{\mathrm{L}}{2} \|\overline{\boldsymbol{\theta}} - \boldsymbol{\theta}\|^2, \quad (\boldsymbol{\theta}, \overline{\boldsymbol{\theta}}) \in \Theta^2 \ . \tag{11}$$

It is easily checked that the quadratic function $\widehat{\mathcal{L}}_i(\boldsymbol{\theta}; \overline{\boldsymbol{\theta}})$ satisfies H1, H2. To compute the gradient $\nabla \mathcal{L}_i(\overline{\boldsymbol{\theta}})$, we apply the re-parametrization technique suggested in (Paisley et al., 2012; Kingma & Welling, 2014; Blundell et al., 2015). Let $t : \mathbb{R}^d \times \Theta \mapsto \mathbb{R}^d$ be a differentiable function *w.r.t.* $\boldsymbol{\theta} \in \Theta$ which is designed such that the law of $w = t(z, \overline{\boldsymbol{\theta}})$ is $q(\cdot, \overline{\boldsymbol{\theta}})$, where $z \sim \mathcal{N}_d(0, \mathbf{I})$. By (Blundell et al., 2015, Proposition 1), the gradient of $-r_i(\cdot)$ in (10) is:

$$\nabla_{\boldsymbol{\theta}} \mathbb{E}_{q(w;\overline{\boldsymbol{\theta}})} \left[ \log p(y_i|x_i, w) \right] = \mathbb{E}_{z \sim \mathcal{N}_d(0,\mathbf{I})} \left[ \mathrm{J}^t_{\boldsymbol{\theta}}(z, \overline{\boldsymbol{\theta}}) \nabla_w \log p(y_i|x_i, w) \big|_{w=t(z,\overline{\boldsymbol{\theta}})} \right] , \tag{12}$$

where for each $z \in \mathbb{R}^d$, $\mathrm{J}^t_{\boldsymbol{\theta}}(z, \overline{\boldsymbol{\theta}})$ is the Jacobian of the function $t(z, \cdot)$ with respect to $\boldsymbol{\theta}$ evaluated at $\overline{\boldsymbol{\theta}}$. In addition, for most cases, the term $\nabla d(\overline{\boldsymbol{\theta}})$ can be evaluated in closed form as the gradient of the KL between the prior distribution $\pi(\cdot)$ and the variational candidate $q(\cdot, \boldsymbol{\theta})$.

$$r_i(\boldsymbol{\theta}; \overline{\boldsymbol{\theta}}, z) := \left\langle \nabla_{\boldsymbol{\theta}} d(\overline{\boldsymbol{\theta}}) - \mathrm{J}^t_{\boldsymbol{\theta}}(z, \overline{\boldsymbol{\theta}}) \nabla_w \log p(y_i|x_i, w) \big|_{w=t(z,\overline{\boldsymbol{\theta}})} \, | \, \boldsymbol{\theta} - \overline{\boldsymbol{\theta}} \right\rangle + \frac{L}{2} \|\boldsymbol{\theta} - \overline{\boldsymbol{\theta}}\|^2 \ . \tag{13}$$

Finally, using (11) and (13), the surrogate function (6) is given by $\widetilde{\mathcal{L}}_i(\boldsymbol{\theta}; \overline{\boldsymbol{\theta}}, \{z_m\}_{m=1}^M) := M^{-1} \sum_{m=1}^M r_i(\boldsymbol{\theta}; \overline{\boldsymbol{\theta}}, z_m)$ where $\{z_m\}_{m=1}^M$ are i.i.d samples drawn from $\mathcal{N}(0, \mathbf{I})$.

## 3 CONVERGENCE ANALYSIS

We now provide asymptotic and non-asymptotic convergence results of our method. Assume:

**H3.** *For all $i \in [\![1, n]\!]$, $\overline{\boldsymbol{\theta}} \in \Theta$, $z_i \in \mathsf{Z}$, $r_i(\cdot; \overline{\boldsymbol{\theta}}, z_i)$ is convex on $\Theta$ and is lower bounded.*

We are particularly interested in the *constrained optimization* setting where $\Theta$ is a bounded set. To this end, we control the supremum norm of the MC approximation, introduced in (6), as:

**H4.** *For the samples $\{z_{i,m}\}_{m=1}^M$, there exist finite constants $C_\mathsf{r}$ and $C_\mathsf{gr}$ such that*

$$C_\mathsf{r} := \sup_{\overline{\boldsymbol{\theta}} \in \Theta} \sup_{M > 0} \frac{1}{\sqrt{M}} \mathbb{E}_{\overline{\boldsymbol{\theta}}} \left[ \sup_{\boldsymbol{\theta} \in \Theta} \left| \sum_{m=1}^M \left\{ r_i(\boldsymbol{\theta}; \overline{\boldsymbol{\theta}}, z_{i,m}) - \widehat{\mathcal{L}}_i(\boldsymbol{\theta}; \overline{\boldsymbol{\theta}}) \right\} \right| \right]$$

$$C_\mathsf{gr} := \sup_{\overline{\boldsymbol{\theta}} \in \Theta} \sup_{M > 0} \sqrt{M} \mathbb{E}_{\overline{\boldsymbol{\theta}}} \left[ \sup_{\boldsymbol{\theta} \in \Theta} \left| \frac{1}{M} \sum_{m=1}^M \frac{\widehat{\mathcal{L}}'_i(\boldsymbol{\theta}, \boldsymbol{\theta} - \overline{\boldsymbol{\theta}}; \overline{\boldsymbol{\theta}}) - r'_i(\boldsymbol{\theta}, \boldsymbol{\theta} - \overline{\boldsymbol{\theta}}; \overline{\boldsymbol{\theta}}, z_{i,m})}{\|\overline{\boldsymbol{\theta}} - \boldsymbol{\theta}\|} \right|^2 \right]$$

*for all $i \in [\![1, n]\!]$, and we denoted by $\mathbb{E}_{\overline{\boldsymbol{\theta}}}[\cdot]$ the expectation w.r.t. a Markov chain $\{z_{i,m}\}_{m=1}^M$ with initial distribution $\xi_i(\cdot; \overline{\boldsymbol{\theta}})$, transition kernel $\Pi_{i,\overline{\boldsymbol{\theta}}}$, and stationary distribution $p_i(\cdot; \overline{\boldsymbol{\theta}})$.*

**Some intuitions behind the controlling terms:** It is common in statistical and optimization problems, to deal with the manipulation and the control of random variables indexed by sets with an infinite number of elements. Here, the controlled random variable is an image of a continuous function defined as $r_i(\boldsymbol{\theta}; \overline{\boldsymbol{\theta}}, z_{i,m}) - \widehat{\mathcal{L}}_i(\boldsymbol{\theta}; \overline{\boldsymbol{\theta}})$ for all $z \in \mathsf{Z}$ and for fixed $(\boldsymbol{\theta}, \overline{\boldsymbol{\theta}}) \in \Theta^2$. To characterize such control, we will have recourse to the notion of metric entropy (or bracketing number) as developed in (Van der Vaart, 2000; Vershynin, 2018; Wainwright, 2019). A collection of results from those references gives intuition behind our assumption H4, which is classical in empirical processes. In (Vershynin, 2018, Theorem 8.2.3), the authors recall the uniform law of large numbers:

$$\mathbb{E} \left[ \sup_{f \in \mathcal{F}} \left| \frac{1}{M} \sum_{i=1}^M f(z_{i,m}) - \mathbb{E}[f(z_i)] \right| \right] \leq \frac{CL}{\sqrt{M}} \quad \text{for all} \quad z_{i,m}, i \in [\![1, M]\!] \ ,$$

where $\mathcal{F}$ is a class of $L$-Lipschitz functions. Moreover, in (Vershynin, 2018, Theorem 8.1.3 ) and (Wainwright, 2019, Theorem 5.22), the application of the Dudley inequality yields:

$$\mathbb{E}[\sup_{f \in \mathcal{F}} |X_f - X_0|] \leq \frac{1}{\sqrt{M}} \int_0^1 \sqrt{\log \mathcal{N}(\mathcal{F}, \|\cdot\|_\infty, \varepsilon)} d\varepsilon \ ,$$

where $\mathcal{N}\left(\mathcal{F}, \|\cdot\|_\infty, \varepsilon\right)$ is the bracketing number and $\epsilon$ denotes the level of approximation (the bracketing number goes to infinity when $\epsilon \to 0$). Finally, in (Van der Vaart, 2000, p.271, Example), $\mathcal{N}\left(\mathcal{F}, \|\cdot\|_\infty, \varepsilon\right)$ is bounded from above for a class of parametric functions $\mathcal{F} = f_{\boldsymbol{\theta}} : \boldsymbol{\theta} \in \Theta$:

$$\mathcal{N}\left(\mathcal{F}, \|\cdot\|_\infty, \varepsilon\right) \leq K \left(\frac{\operatorname{diam}\Theta}{\varepsilon}\right)^d, \quad \text{for all} \quad 0 < \varepsilon < \operatorname{diam}\Theta \,.$$

The authors acknowledge that those bounds are a dramatic manifestation of the curse of dimensionality happening when sampling is needed. Nevertheless, the dependence on the dimension highly depends on the class of surrogate functions $\mathcal{F}$ used in our scheme, as smaller bounds on these controlling terms can be derived for simpler class of functions, such as quadratic functions.

**Stationarity measure.** As problem (1) is a constrained optimization task, we consider the following stationarity measure:

$$g(\overline{\boldsymbol{\theta}}) := \inf_{\boldsymbol{\theta}\in\Theta} \frac{\mathcal{L}'(\overline{\boldsymbol{\theta}}, \boldsymbol{\theta} - \overline{\boldsymbol{\theta}})}{\|\overline{\boldsymbol{\theta}} - \boldsymbol{\theta}\|} \quad \text{and} \quad g(\overline{\boldsymbol{\theta}}) = g_+(\overline{\boldsymbol{\theta}}) - g_-(\overline{\boldsymbol{\theta}}) \,, \tag{14}$$

where $g_+(\overline{\boldsymbol{\theta}}) := \max\{0, g(\overline{\boldsymbol{\theta}})\}$, $g_-(\overline{\boldsymbol{\theta}}) := -\min\{0, g(\overline{\boldsymbol{\theta}})\}$ denote the positive and negative part of $g(\overline{\boldsymbol{\theta}})$, respectively. Note that $\overline{\boldsymbol{\theta}}$ is a stationary point if and only if $g_-(\overline{\boldsymbol{\theta}}) = 0$ (Fletcher et al., 2002). Furthermore, suppose that the sequence $\{\boldsymbol{\theta}^{(k)}\}_{k\geq 0}$ has a limit point $\overline{\boldsymbol{\theta}}$ that is a stationary point, then one has $\lim_{k\to\infty} g_-(\boldsymbol{\theta}^{(k)}) = 0$. Thus, the sequence $\{\boldsymbol{\theta}^{(k)}\}_{k\geq 0}$ is said to satisfy an *asymptotic stationary point condition*. This is equivalent to (Mairal, 2015, Definition 2.4).

To facilitate our analysis, we define $\tau_i^k$ as the iteration index where the $i$-th function is last accessed in the MISSO method prior to iteration $k$, $\tau_{i_k}^{k+1} = k$ for instance. We define:

$$\widehat{\mathcal{L}}^{(k)}(\boldsymbol{\theta}) := \tfrac{1}{n}\sum_{i=1}^n \widehat{\mathcal{L}}_i(\boldsymbol{\theta}; \boldsymbol{\theta}^{(\tau_i^k)}), \quad \widehat{e}^{(k)}(\boldsymbol{\theta}) := \widehat{\mathcal{L}}^{(k)}(\boldsymbol{\theta}) - \mathcal{L}(\boldsymbol{\theta}), \quad \overline{M}_{(k)} := \sum_{k=0}^{K_{\max}-1} M_{(k)}^{-1/2} \,. \tag{15}$$

We first establish a non-asymptotic convergence rate for the MISSO method:

---

**Theorem 1.** *Under H1-H4. For any $K_{\max} \in \mathbb{N}$, let $K$ be an independent discrete r.v. drawn uniformly from $\{0, ..., K_{\max} - 1\}$ and define the following quantity:*

$$\Delta_{(K_{\max})} := 2nL\mathbb{E}[\widetilde{\mathcal{L}}^{(0)}(\boldsymbol{\theta}^{(0)}) - \widetilde{\mathcal{L}}^{(K_{\max})}(\boldsymbol{\theta}^{(K_{\max})})] + 4LC_{\mathsf{r}}\overline{M}_{(k)} \,.$$

*Then we have following non-asymptotic bounds:*

$$\mathbb{E}\big[\|\nabla\widehat{e}^{(K)}(\boldsymbol{\theta}^{(K)})\|^2\big] \leq \frac{\Delta_{(K_{\max})}}{K_{\max}} \quad \text{and} \quad \mathbb{E}[g_-(\boldsymbol{\theta}^{(K)})] \leq \sqrt{\frac{\Delta_{(K_{\max})}}{K_{\max}}} + \frac{C_{\mathsf{gr}}}{K_{\max}}\overline{M}_{(k)} \,. \tag{16}$$

---

Note that $\Delta_{(K_{\max})}$ is finite for any $K_{\max} \in \mathbb{N}$.

**Iteration Complexity of MISSO.** As expected, the MISSO method converges to a stationary point of (1) asymptotically and at a sublinear rate $\mathbb{E}[g_-^{(K)}] \leq \mathcal{O}(\sqrt{\Delta_{(K_{\max})}/K_{\max}})$. In other terms, MISSO requires $\mathcal{O}(nL/\epsilon)$ iterations to reach an $\epsilon$-stationary point when the suboptimality condition, that characterizes stationarity, is $\mathbb{E}\big[\|g_-(\boldsymbol{\theta}^{(K)})\|^2\big]$. Note that this stationarity criterion are similar to the usual quantity used in stochastic nonconvex optimization, *i.e.*, $\mathbb{E}\big[\|\nabla\mathcal{L}(\boldsymbol{\theta}^{(K)})\|^2\big]$. In fact, when the optimization problem (1) is unconstrained, *i.e.*, $\Theta = \mathbb{R}^p$, then $\mathbb{E}\big[g(\boldsymbol{\theta}^{(K)})\big] = \mathbb{E}\big[\nabla\mathcal{L}(\boldsymbol{\theta}^{(K)})\big]$.

**Sample Complexity of MISSO.** Regarding the sample complexity of our method, setting $M_{(k)} = k^2/n^2$, as a non-decreasing sequence of integers satisfying $\sum_{k=0}^\infty M_{(k)}^{-1/2} < \infty$, in order to keep $\Delta_{(K_{\max})} \asymp nL$, then the MISSO method requires $\sum_{k=0}^{nL/\epsilon} k^2/n^2 = nL^3/\epsilon^3$ samples to reach an $\epsilon$-stationary point.

Furthermore, we remark that the MISO method can be analyzed in Theorem 1 as a special case of the MISSO method satisfying $C_{\mathsf{r}} = C_{\mathsf{gr}} = 0$. In this case, while the asymptotic convergence is well known from (Mairal, 2015) [cf. H4], Eq. (16) gives a non-asymptotic rate of $\mathbb{E}[g_-^{(K)}] \leq$

$\mathcal{O}(\sqrt{nL/K_{\max}})$ which is new to our best knowledge. Next, we show that under an additional assumption on the sequence of batch size $M_{(k)}$, the MISSO method converges almost surely to a stationary point:

---

**Theorem 2.** *Under H1-H4. In addition, assume that $\{M_{(k)}\}_{k \geq 0}$ is a non-decreasing sequence of integers which satisfies $\sum_{k=0}^{\infty} M_{(k)}^{-1/2} < \infty$. Then:*

1. *the negative part of the stationarity measure converges a.s. to zero, i.e., $\lim\limits_{k \to \infty} g_{-}(\boldsymbol{\theta}^{(k)}) \overset{a.s.}{=} 0$.*

2. *the objective value $\mathcal{L}(\boldsymbol{\theta}^{(k)})$ converges a.s. to a finite number $\underline{\mathcal{L}}$, i.e., $\lim_{k \to \infty} \mathcal{L}(\boldsymbol{\theta}^{(k)}) \overset{a.s.}{=} \underline{\mathcal{L}}$.*

---

In particular, the first result above shows that the sequence $\{\boldsymbol{\theta}^{(k)}\}_{k \geq 0}$ produced by the MISSO method satisfies an *asymptotic stationary point condition*.

## 4 NUMERICAL EXPERIMENTS

### 4.1 BINARY LOGISTIC REGRESSION WITH MISSING VALUES

This application follows **Example 1** described in Section 2. We consider a binary regression setup, $((y_i, z_i), i \in [\![n]\!])$ where $y_i \in \{0, 1\}$ is a binary response and $z_i = (z_{i,j} \in \mathbb{R}, j \in [\![p]\!])$ is a covariate vector. The vector of covariates $z_i = [z_{i,\mathrm{mis}}, z_{i,\mathrm{obs}}]$ is not fully observed where we denote by $z_{i,\mathrm{mis}}$ the missing values and $z_{i,\mathrm{obs}}$ the observed covariate. It is assumed that $(z_i, i \in [\![n]\!])$ are i.i.d. and marginally distributed according to $\mathcal{N}(\boldsymbol{\beta}, \boldsymbol{\Omega})$ where $\beta \in \mathbb{R}^p$ and $\Omega$ is a positive definite $p \times p$ matrix. We define the conditional distribution of the observations $y_i$ given $z_i = (z_{i,\mathrm{mis}}, z_{i,\mathrm{obs}})$ as:

$$p_i(y_i|z_i) = S(\boldsymbol{\delta}^\top \bar{z}_i)^{y_i} \left(1 - S(\boldsymbol{\delta}^\top \bar{z}_i)\right)^{1-y_i} , \tag{17}$$

where for $u \in \mathbb{R}$, $S(u) = 1/(1+\mathrm{e}^{-u})$, $\boldsymbol{\delta} = (\delta_0, \cdots, \delta_p)$ are the logistic parameters and $\bar{z}_i = (1, z_i)$. Here, $\boldsymbol{\theta} = (\boldsymbol{\delta}, \boldsymbol{\beta}, \boldsymbol{\Omega})$ is the parameter to estimate. For $i \in [\![n]\!]$, the complete log-likelihood reads:

$$\log f_i(z_{i,\mathrm{mis}}, \boldsymbol{\theta}) \propto y_i \boldsymbol{\delta}^\top \bar{z}_i - \log\left(1 + \exp(\boldsymbol{\delta}^\top \bar{z}_i)\right) - \frac{1}{2}\log(|\boldsymbol{\Omega}|) + \frac{1}{2}\mathrm{Tr}\left(\boldsymbol{\Omega}^{-1}(z_i - \boldsymbol{\beta})(z_i - \boldsymbol{\beta})^\top\right).$$

**Fitting a logistic regression model on the TraumaBase dataset:** We apply the MISSO method to fit a logistic regression model on the TraumaBase (http://traumabase.eu) dataset, which consists of data collected from 15 trauma centers in France, covering measurements on patients from the initial to last stage of trauma. This dataset includes information from the first stage of the trauma, namely initial observations on the patient's accident site to the last stage being intense care at the hospital and counts more than 200 variables measured for more than 7 000 patients. Since the dataset considered is heterogeneous – coming from multiple sources with frequently missed entries – we apply the latent data model described in (17) to *predict the risk of a severe hemorrhage* which is one of the main cause of death after a major trauma.

Similar to (Jiang et al., 2018), we select $p = 16$ influential quantitative measurements, on $n = 6384$ patients. For the Monte Carlo sampling of $z_{i,\mathrm{mis}}$, required while running MISSO, we run a Metropolis-Hastings algorithm with the target distribution $p(\cdot|z_{i,\mathrm{obs}}, y_i; \boldsymbol{\theta}^{(k)})$.

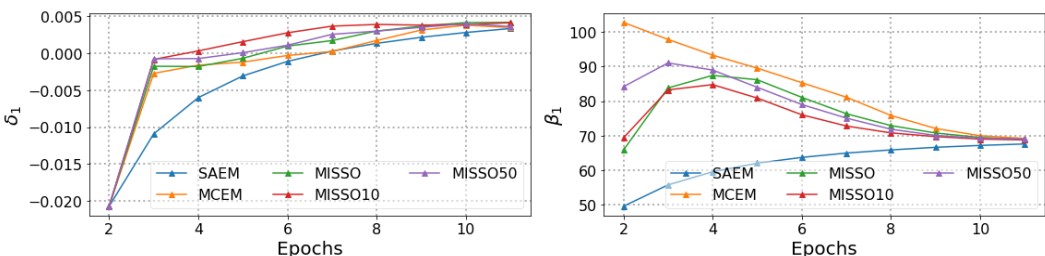

Figure 1: Convergence of parameters $\boldsymbol{\delta}$ and $\boldsymbol{\beta}$ for the SAEM, the MCEM and the MISSO methods. The convergence is plotted against number of passes over the data.

We compare in Figure 1 the convergence behavior of the estimated parameters $\boldsymbol{\delta}$ and $\boldsymbol{\beta}$ using SAEM (Delyon et al., 1999) (with stepsize $\gamma_k = 1/k^\alpha$ where $\alpha = 0.6$ after tuning), MCEM (Wei

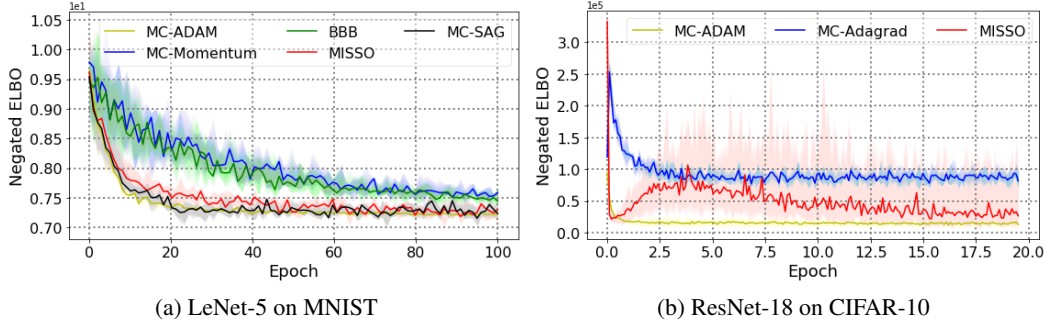

Figure 2: Negated ELBO versus epochs elapsed for fitting (a) Bayesian LeNet-5 on MNIST and (b) Bayesian ResNet-18 on CIFAR-10. The solid curve is obtained from averaging over 5 independent runs of the methods, and the shaded area represents the standard deviation.

& Tanner, 1990) and the proposed MISSO method. For the MISSO method, we set the batch size to $M_{(k)} = 10 + k^2$ and we examine with selecting different number of functions in Line 5 in the method – the default settings with 1 (MISSO), 10% (MISSO10) and 50% (MISSO50) minibatches per iteration. From Figure 1, the MISSO method converges to a static value with less number of epochs than the MCEM, SAEM methods. It is worth noting that the difference among the MISSO runs for different number of selected functions demonstrates a variance-cost tradeoff. Though wall clock times are similar for all methods, they are reported in the appendix for completeness.

## 4.2 TRAINING BAYESIAN CNN USING MISSO

This application follows **Example 2** described in Section 2. We use variational inference and the ELBO loss (10) to fit Bayesian Neural Networks on different datasets. At iteration $k$, minimizing the sum of stochastic surrogates defined as in (6) and (13) yields the following MISSO update — step (i) pick a function index $i_k$ uniformly on $[\![n]\!]$; step (ii) sample a Monte Carlo batch $\{z_m^{(k)}\}_{m=1}^{M_{(k)}}$ from $\mathcal{N}(0, \mathbf{I})$; and step (iii) update the parameters, with $\tilde{w} = t(\boldsymbol{\theta}^{(k-1)}, z_m^{(k)})$, as

$$\mu_\ell^{(k)} = \hat{\mu}_\ell^{(\tau^k)} - \frac{\gamma}{n} \sum_{i=1}^n \hat{\boldsymbol{\delta}}_{\mu_\ell, i}^{(k)} \,,$$

$$\hat{\boldsymbol{\delta}}_{\mu_\ell, i_k}^{(k)} = -\frac{1}{M_{(k)}} \sum_{m=1}^{M_{(k)}} \nabla_w \log p(y_{i_k} | x_{i_k}, \tilde{w}) + \nabla_{\mu_\ell} d(\boldsymbol{\theta}^{(k-1)}) \,,$$

where $\hat{\mu}_\ell^{(\tau^k)} = \frac{1}{n} \sum_{i=1}^n \mu_\ell^{(\tau_i^k)}$ and $d(\boldsymbol{\theta}) = n^{-1} \sum_{\ell=1}^d \left( -\log(\sigma) + (\sigma^2 + \mu_\ell^2)/2 - 1/2 \right)$.

**Bayesian LeNet-5 on MNIST (LeCun et al., 1998):** We apply the MISSO method to fit a Bayesian variant of LeNet-5 (LeCun et al., 1998). We train this network on the MNIST dataset (LeCun, 1998). The training set is composed of $n = 55\,000$ handwritten digits, $28 \times 28$ images. Each image is labelled with its corresponding number (from zero to nine). Under the prior distribution $\pi$, see (8), the weights are assumed independent and identically distributed according to $\mathcal{N}(0, 1)$. We also assume that $q(\cdot; \boldsymbol{\theta}) \equiv \mathcal{N}(\mu, \sigma^2 \mathbf{I})$. The variational posterior parameters are thus $\boldsymbol{\theta} = (\mu, \sigma)$ where $\mu = (\mu_\ell, \ell \in [\![d]\!])$ where $d$ is the number of weights in the neural network. We use the re-parametrization as $w = t(\boldsymbol{\theta}, z) = \mu + \sigma z$ with $z \sim \mathcal{N}(0, \mathbf{I})$.

**Bayesian ResNet-18 (He et al., 2016) on CIFAR-10 (Krizhevsky et al., 2012):** We train here the Bayesian variant of the ResNet-18 neural network introduced in (He et al., 2016) on CIFAR-10. The latter dataset is composed of $n = 60\,000$ handwritten digits, $32 \times 32$ colour images in 10 classes, with $6\,000$ images per class. As in the previous example, the weights are assumed independent and identically distributed according to $\mathcal{N}(0, \mathbf{I})$. Standard hyperparameters values found in the literature, such as the annealing constant or the number of MC samples, were used for the benchmark methods. For efficiency purpose and lower variance, the Flipout estimator (Wen et al., 2018) is used.

**Experiment Results:** We compare the convergence of the *Monte Carlo variants* of the following state of the art optimization algorithms — the ADAM (Kingma & Ba, 2015), the Momentum (Sutskever et al., 2013) and the SAG (Schmidt et al., 2017) methods versus the *Bayes by Backprop* (BBB) (Blundell et al., 2015) and our proposed MISSO method. For all these methods, the loss function (10) and its gradients were computed by Monte Carlo integration based on the reparametrization described above. The mini-batch of indices and MC samples are respectively set to 128 and $M_{(k)} = k$. The learning rates are set to $10^{-3}$ for LeNet-5 and $10^{-4}$ for Resnet-18.

Figure 2(a) shows the convergence of the negated evidence lower bound against the number of passes over data (one pass represents an epoch). As observed, the proposed MISSO method outperforms *Bayes by Backprop* and Momentum, while similar convergence rates are observed with the MISSO, ADAM and SAG methods for our experiment on MNIST dataset using a Bayesian variant of LeNet-5. On the other hand, the experiment conducted on CIFAR-10 (Figure 2(b)) using a much larger network, *i.e.,* a Bayesian variant of ResNet-18 showcases the need of a well-tuned adaptive methods to reach lower training loss (and also faster). Our MISSO method is similar to the Monte Carlo variant of ADAM but slower than Adagrad optimizer. Recall that the purpose of this paper is to provide a common class of optimizers, such as VI, in order to study their convergence behaviors, and not to introduce a novel method outperforming the baselines methods. We report wall clock times for all methods in the appendix for completeness.

## 5 CONCLUSION

We present a unifying framework for minimizing a nonconvex and nonsmooth finite-sum objective function using incremental surrogates when the latter functions are expressed as an expectation and are intractable. Our approach covers a large class of nonconvex applications in machine learning such as logistic regression with missing values and variational inference. We provide both finite-time and asymptotic guarantees of our incremental stochastic surrogate optimization technique and illustrate our findings training a binary logistic regression with missing covariates to predict hemorrhagic shock and Bayesian variants of two Convolutional Neural Networks on benchmark datasets.

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

# A PROOFS OF THE THEORETICAL RESULTS

## A.1 PROOF OF THEOREM 1

**Theorem.** *Under H1-H4. For any $K_{\max} \in \mathbb{N}$, let $K$ be an independent discrete r.v. drawn uniformly from $\{0, ..., K_{\max} - 1\}$ and define the following quantity:*

$$\Delta_{(K_{\max})} := 2nL\mathbb{E}[\widetilde{\mathcal{L}}^{(0)}(\boldsymbol{\theta}^{(0)}) - \widetilde{\mathcal{L}}^{(K_{\max})}(\boldsymbol{\theta}^{(K_{\max})})] + 4LC_{\mathsf{r}}\overline{M}_{(k)} \ .$$

*Then we have following non-asymptotic bounds:*

$$\mathbb{E}\big[\|\nabla\widehat{e}^{(K)}(\boldsymbol{\theta}^{(K)})\|^2\big] \leq \frac{\Delta_{(K_{\max})}}{K_{\max}} \quad and \quad \mathbb{E}[g_-(\boldsymbol{\theta}^{(K)})] \leq \sqrt{\frac{\Delta_{(K_{\max})}}{K_{\max}}} + \frac{C_{\mathsf{gr}}}{K_{\max}}\overline{M}_{(k)} \ .$$

**Proof** We begin by recalling the definition

$$\widetilde{\mathcal{L}}^{(k)}(\boldsymbol{\theta}) := \frac{1}{n}\sum_{i=1}^n \widetilde{\mathcal{A}}_i^k(\boldsymbol{\theta}) \ .$$

Notice that

$$\widetilde{\mathcal{L}}^{(k+1)}(\boldsymbol{\theta}) = \frac{1}{n}\sum_{i=1}^n \widetilde{\mathcal{L}}_i(\boldsymbol{\theta}; \boldsymbol{\theta}^{(\tau_i^{k+1})}, \{z_{i,m}^{(\tau_i^{k+1})}\}_{m=1}^{M_{(\tau_i^{k+1})}})$$

$$= \widetilde{\mathcal{L}}^{(k)}(\boldsymbol{\theta}) + \frac{1}{n}\big(\widetilde{\mathcal{L}}_{i_k}(\boldsymbol{\theta}; \boldsymbol{\theta}^{(k)}, \{z_{i_k,m}^{(k)}\}_{m=1}^{M_{(k)}}) - \widetilde{\mathcal{L}}_{i_k}(\boldsymbol{\theta}; \boldsymbol{\theta}^{(\tau_{i_k}^k)}, \{z_{i_k,m}^{(\tau_{i_k}^k)}\}_{m=1}^{M_{(\tau_{i_k}^k)}})\big) \ .$$

Furthermore, we recall that

$$\widehat{\mathcal{L}}^{(k)}(\boldsymbol{\theta}) := \tfrac{1}{n}\sum_{i=1}^n \widehat{\mathcal{L}}_i(\boldsymbol{\theta}; \boldsymbol{\theta}^{(\tau_i^k)}), \quad \widehat{e}^{(k)}(\boldsymbol{\theta}) := \widehat{\mathcal{L}}^{(k)}(\boldsymbol{\theta}) - \mathcal{L}(\boldsymbol{\theta}) \ .$$

Due to H2, we have

$$\|\nabla\widehat{e}^{(k)}(\boldsymbol{\theta}^{(k)})\|^2 \leq 2L\widehat{e}^{(k)}(\boldsymbol{\theta}^{(k)}) \ . \tag{18}$$

To prove the first bound in (16), using the optimality of $\boldsymbol{\theta}^{(k+1)}$, one has

$$\widetilde{\mathcal{L}}^{(k+1)}(\boldsymbol{\theta}^{(k+1)}) \leq \widetilde{\mathcal{L}}^{(k+1)}(\boldsymbol{\theta}^{(k)})$$
$$= \widetilde{\mathcal{L}}^{(k)}(\boldsymbol{\theta}^{(k)}) + \tfrac{1}{n}\big(\widetilde{\mathcal{L}}_{i_k}(\boldsymbol{\theta}^{(k)}; \boldsymbol{\theta}^{(k)}, \{z_{i_k,m}^{(k)}\}_{m=1}^{M_{(k)}}) - \widetilde{\mathcal{L}}_{i_k}(\boldsymbol{\theta}^{(k)}; \boldsymbol{\theta}^{(\tau_{i_k}^k)}, \{z_{i_k,m}^{(\tau_{i_k}^k)}\}_{m=1}^{M_{(\tau_{i_k}^k)}})\big) \ . \tag{19}$$

Let $\mathcal{F}_k$ be the filtration of random variables up to iteration $k$, *i.e.*, $\{i_{\ell-1}, \{z_{i_{\ell-1},m}^{(\ell-1)}\}_{m=1}^{M_{(\ell-1)}}, \boldsymbol{\theta}^{(\ell)}\}_{\ell=1}^k$. We observe that the conditional expectation evaluates to

$$\mathbb{E}_{i_k}\big[\mathbb{E}\big[\widetilde{\mathcal{L}}_{i_k}(\boldsymbol{\theta}^{(k)}; \boldsymbol{\theta}^{(k)}, \{z_{i_k,m}^{(k)}\}_{m=1}^{M_{(k)}})|\mathcal{F}_k, i_k\big]|\mathcal{F}_k\big]$$

$$= \mathcal{L}(\boldsymbol{\theta}^{(k)}) + \mathbb{E}_{i_k}\big[\mathbb{E}\big[\frac{1}{M_{(k)}}\sum_{m=1}^{M_{(k)}} r_{i_k}(\boldsymbol{\theta}^{(k)}; \boldsymbol{\theta}^{(k)}, z_{i_k,m}^{(k)}) - \widehat{\mathcal{L}}_{i_k}(\boldsymbol{\theta}^{(k)}; \boldsymbol{\theta}^{(k)})|\mathcal{F}_k, i_k\big]|\mathcal{F}_k\big]$$

$$\leq \mathcal{L}(\boldsymbol{\theta}^{(k)}) + \frac{C_{\mathsf{r}}}{\sqrt{M_{(k)}}} \ ,$$

where the last inequality is due to H4. Moreover,

$$\mathbb{E}\big[\widetilde{\mathcal{L}}_{i_k}(\boldsymbol{\theta}^{(k)}; \boldsymbol{\theta}^{(\tau_{i_k}^k)}, \{z_{i_k,m}^{(\tau_{i_k}^k)}\}_{m=1}^{M_{(\tau_{i_k}^k)}})|\mathcal{F}_k\big] = \frac{1}{n}\sum_{i=1}^n \widetilde{\mathcal{L}}_i(\boldsymbol{\theta}^{(k)}; \boldsymbol{\theta}^{(\tau_i^k)}, \{z_{i,m}^{(\tau_i^k)}\}_{m=1}^{M_{(\tau_i^k)}}) = \widetilde{\mathcal{L}}^{(k)}(\boldsymbol{\theta}^{(k)}) \ .$$

Taking the conditional expectations on both sides of (19) and re-arranging terms give:

$$\widetilde{\mathcal{L}}^{(k)}(\boldsymbol{\theta}^{(k)}) - \mathcal{L}(\boldsymbol{\theta}^{(k)}) \leq n\mathbb{E}\big[\widetilde{\mathcal{L}}^{(k)}(\boldsymbol{\theta}^{(k)}) - \widetilde{\mathcal{L}}^{(k+1)}(\boldsymbol{\theta}^{(k+1)})|\mathcal{F}_k\big] + \frac{C_{\mathsf{r}}}{\sqrt{M_{(k)}}} \ . \tag{20}$$

Proceeding from (20), we observe the following lower bound for the left hand side

$$
\widetilde{\mathcal{L}}^{(k)}(\boldsymbol{\theta}^{(k)}) - \mathcal{L}(\boldsymbol{\theta}^{(k)}) \overset{(a)}{=} \widetilde{\mathcal{L}}^{(k)}(\boldsymbol{\theta}^{(k)}) - \widehat{\mathcal{L}}^{(k)}(\boldsymbol{\theta}^{(k)}) + \widehat{e}^{(k)}(\boldsymbol{\theta}^{(k)})
$$

$$
\overset{(b)}{\geq} \widetilde{\mathcal{L}}^{(k)}(\boldsymbol{\theta}^{(k)}) - \widehat{\mathcal{L}}^{(k)}(\boldsymbol{\theta}^{(k)}) + \frac{1}{2L}\|\nabla\widehat{e}^{(k)}(\boldsymbol{\theta}^{(k)})\|^2
$$

$$
= \underbrace{\frac{1}{n}\sum_{i=1}^{n}\left\{\frac{1}{M_{(\tau_i^k)}}\sum_{m=1}^{M_{(\tau_i^k)}} r_i(\boldsymbol{\theta}^{(k)};\boldsymbol{\theta}^{(\tau_i^k)}, z_{i,m}^{(\tau_i^k)}) - \widehat{\mathcal{L}}_i(\boldsymbol{\theta}^{(k)};\boldsymbol{\theta}^{(\tau_i^k)})\right\}}_{:=-\delta^{(k)}(\boldsymbol{\theta}^{(k)})} + \frac{1}{2L}\|\nabla\widehat{e}^{(k)}(\boldsymbol{\theta}^{(k)})\|^2 ,
$$

where (a) is due to $\widehat{e}^{(k)}(\boldsymbol{\theta}^{(k)}) = 0$ [cf. H1], (b) is due to (18) and we have defined the summation in the last equality as $-\delta^{(k)}(\boldsymbol{\theta}^{(k)})$. Substituting the above into (20) yields

$$
\frac{\|\nabla\widehat{e}^{(k)}(\boldsymbol{\theta}^{(k)})\|^2}{2L} \leq n\mathbb{E}\big[\widetilde{\mathcal{L}}^{(k)}(\boldsymbol{\theta}^{(k)}) - \widetilde{\mathcal{L}}^{(k+1)}(\boldsymbol{\theta}^{(k+1)})|\mathcal{F}_k\big] + \frac{C_{\mathsf{r}}}{\sqrt{M_{(k)}}} + \delta^{(k)}(\boldsymbol{\theta}^{(k)}) . \tag{21}
$$

Observe the following upper bound on the total expectations:

$$
\mathbb{E}\big[\delta^{(k)}(\boldsymbol{\theta}^{(k)})\big] \leq \mathbb{E}\Big[\frac{1}{n}\sum_{i=1}^{n}\frac{C_{\mathsf{r}}}{\sqrt{M_{(\tau_i^k)}}}\Big] ,
$$

which is due to H4. It yields

$$
\mathbb{E}\big[\|\nabla\widehat{e}^{(k)}(\boldsymbol{\theta}^{(k)})\|^2\big] \leq 2nL\,\mathbb{E}\big[\widetilde{\mathcal{L}}^{(k)}(\boldsymbol{\theta}^{(k)}) - \widetilde{\mathcal{L}}^{(k+1)}(\boldsymbol{\theta}^{(k+1)})\big] + \frac{2LC_{\mathsf{r}}}{\sqrt{M_{(k)}}} + \frac{1}{n}\sum_{i=1}^{n}\mathbb{E}\Big[\frac{2LC_{\mathsf{r}}}{\sqrt{M_{(\tau_i^k)}}}\Big] .
$$

Finally, for any $K_{\mathsf{max}} \in \mathbb{N}$, we let $K$ be a discrete r.v. that is uniformly drawn from $\{0, 1, ..., K_{\mathsf{max}} - 1\}$. Using H4 and taking total expectations lead to

$$
\mathbb{E}\big[\|\nabla\widehat{e}^{(K)}(\boldsymbol{\theta}^{(K)})\|^2\big] = \frac{1}{K_{\mathsf{max}}}\sum_{k=0}^{K_{\mathsf{max}}-1}\mathbb{E}[\|\nabla\widehat{e}^{(k)}(\boldsymbol{\theta}^{(k)})\|^2]
$$

$$
\leq \frac{2nL\,\mathbb{E}[\widetilde{\mathcal{L}}^{(0)}(\boldsymbol{\theta}^{(0)}) - \widetilde{\mathcal{L}}^{(K_{\mathsf{max}})}(\boldsymbol{\theta}^{(K_{\mathsf{max}})})]}{K_{\mathsf{max}}} + \frac{2LC_{\mathsf{r}}}{K_{\mathsf{max}}}\sum_{k=0}^{K_{\mathsf{max}}-1}\mathbb{E}\Big[\frac{1}{\sqrt{M_{(k)}}} + \frac{1}{n}\sum_{i=1}^{n}\frac{1}{\sqrt{M_{(\tau_i^k)}}}\Big] . \tag{22}
$$

For all $i \in [\![1, n]\!]$, the index $i$ is selected with a probability equal to $\frac{1}{n}$ when conditioned independently on the past. We observe:

$$
\mathbb{E}[M_{(\tau_i^k)}^{-1/2}] = \sum_{j=1}^{k}\frac{1}{n}\left(1 - \frac{1}{n}\right)^{j-1}M_{(k-j)}^{-1/2} \tag{23}
$$

Taking the sum yields:

$$
\sum_{k=0}^{K_{\mathsf{max}}-1}\mathbb{E}[M_{(\tau_i^k)}^{-1/2}] = \sum_{k=0}^{K_{\mathsf{max}}-1}\sum_{j=1}^{k}\frac{1}{n}\left(1 - \frac{1}{n}\right)^{j-1}M_{(k-j)}^{-1/2} = \sum_{k=0}^{K_{\mathsf{max}}-1}\sum_{l=0}^{k-1}\frac{1}{n}\left(1 - \frac{1}{n}\right)^{k-(l+1)}M_{(l)}^{-1/2}
$$

$$
= \sum_{l=0}^{K_{\mathsf{max}}-1}M_{(l)}^{-1/2}\sum_{k=l+1}^{K_{\mathsf{max}}-1}\frac{1}{n}\left(1 - \frac{1}{n}\right)^{k-(l+1)} \leq \sum_{l=0}^{K_{\mathsf{max}}-1}M_{(l)}^{-1/2} ,
$$

$$
\tag{24}
$$

where the last inequality is due to upper bounding the geometric series. Plugging this back into (22) yields

$$
\mathbb{E}\big[\|\nabla\widehat{e}^{(K)}(\boldsymbol{\theta}^{(K)})\|^2\big] = \frac{1}{K_{\mathsf{max}}}\sum_{k=0}^{K_{\mathsf{max}}-1}\mathbb{E}[\|\nabla\widehat{e}^{(k)}(\boldsymbol{\theta}^{(k)})\|^2]
$$

$$
\leq \frac{2nL\,\mathbb{E}[\widetilde{\mathcal{L}}^{(0)}(\boldsymbol{\theta}^{(0)}) - \widetilde{\mathcal{L}}^{(K_{\mathsf{max}})}(\boldsymbol{\theta}^{(K_{\mathsf{max}})})]}{K_{\mathsf{max}}} + \frac{1}{K_{\mathsf{max}}}\sum_{k=0}^{K_{\mathsf{max}}-1}\frac{4LC_{\mathsf{r}}}{\sqrt{M_{(k)}}} = \frac{\Delta_{(K_{\mathsf{max}})}}{K_{\mathsf{max}}} .
$$

This concludes our proof for the first inequality in (16).

To prove the second inequality of (16), we define the shorthand notations $g^{(k)} := g(\boldsymbol{\theta}^{(k)})$, $g_-^{(k)} := -\min\{0, g^{(k)}\}$, $g_+^{(k)} := \max\{0, g^{(k)}\}$. We observe that

$$
\begin{aligned}
g^{(k)} &= \inf_{\boldsymbol{\theta} \in \Theta} \frac{\mathcal{L}'(\boldsymbol{\theta}^{(k)}, \boldsymbol{\theta} - \boldsymbol{\theta}^{(k)})}{\|\boldsymbol{\theta}^{(k)} - \boldsymbol{\theta}\|} \\
&= \inf_{\boldsymbol{\theta} \in \Theta} \left\{ \frac{\frac{1}{n}\sum_{i=1}^n \widehat{\mathcal{L}}_i'(\boldsymbol{\theta}^{(k)}, \boldsymbol{\theta} - \boldsymbol{\theta}^{(k)}; \boldsymbol{\theta}^{(\tau_i^k)})}{\|\boldsymbol{\theta}^{(k)} - \boldsymbol{\theta}\|} - \frac{\langle \nabla \widehat{e}^{(k)}(\boldsymbol{\theta}^{(k)}) \,|\, \boldsymbol{\theta} - \boldsymbol{\theta}^{(k)} \rangle}{\|\boldsymbol{\theta}^{(k)} - \boldsymbol{\theta}\|} \right\} \\
&\geq -\|\nabla \widehat{e}^{(k)}(\boldsymbol{\theta}^{(k)})\| + \inf_{\boldsymbol{\theta} \in \Theta} \frac{\frac{1}{n}\sum_{i=1}^n \widehat{\mathcal{L}}_i'(\boldsymbol{\theta}^{(k)}, \boldsymbol{\theta} - \boldsymbol{\theta}^{(k)}; \boldsymbol{\theta}^{(\tau_i^k)})}{\|\boldsymbol{\theta}^{(k)} - \boldsymbol{\theta}\|} \,,
\end{aligned}
$$

where the last inequality is due to the Cauchy-Schwarz inequality and we have defined $\widehat{\mathcal{L}}_i'(\boldsymbol{\theta}, \boldsymbol{d}; \boldsymbol{\theta}^{(\tau_i^k)})$ as the directional derivative of $\widehat{\mathcal{L}}_i(\cdot; \boldsymbol{\theta}^{(\tau_i^k)})$ at $\boldsymbol{\theta}$ along the direction $\boldsymbol{d}$. Moreover, for any $\boldsymbol{\theta} \in \Theta$,

$$
\frac{1}{n}\sum_{i=1}^n \widehat{\mathcal{L}}_i'(\boldsymbol{\theta}^{(k)}, \boldsymbol{\theta} - \boldsymbol{\theta}^{(k)}; \boldsymbol{\theta}^{(\tau_i^k)})
$$

$$
= \underbrace{\widetilde{\mathcal{L}}^{(k)'}(\boldsymbol{\theta}^{(k)}, \boldsymbol{\theta} - \boldsymbol{\theta}^{(k)})}_{\geq 0} - \widetilde{\mathcal{L}}^{(k)'}(\boldsymbol{\theta}^{(k)}, \boldsymbol{\theta} - \boldsymbol{\theta}^{(k)}) + \frac{1}{n}\sum_{i=1}^n \widehat{\mathcal{L}}_i'(\boldsymbol{\theta}^{(k)}, \boldsymbol{\theta} - \boldsymbol{\theta}^{(k)}; \boldsymbol{\theta}^{(\tau_i^k)})
$$

$$
\geq \frac{1}{n}\sum_{i=1}^n \left\{ \widehat{\mathcal{L}}_i'(\boldsymbol{\theta}^{(k)}, \boldsymbol{\theta} - \boldsymbol{\theta}^{(k)}; \boldsymbol{\theta}^{(\tau_i^k)}) - \frac{1}{M_{(\tau_i^k)}} \sum_{m=1}^{M_{(\tau_i^k)}} r_i'(\boldsymbol{\theta}^{(k)}, \boldsymbol{\theta} - \boldsymbol{\theta}^{(k)}; \boldsymbol{\theta}^{(\tau_i^k)}, z_{i,m}^{(\tau_i^k)}) \right\},
$$

where the inequality is due to the optimality of $\boldsymbol{\theta}^{(k)}$ and the convexity of $\widetilde{\mathcal{L}}^{(k)}(\boldsymbol{\theta})$ [cf. H3]. Denoting a scaled version of the above term as:

$$
\epsilon^{(k)}(\boldsymbol{\theta}) := \frac{\frac{1}{n}\sum_{i=1}^n \left\{ \frac{1}{M_{(\tau_i^k)}} \sum_{m=1}^{M_{(\tau_i^k)}} r_i'(\boldsymbol{\theta}^{(k)}, \boldsymbol{\theta} - \boldsymbol{\theta}^{(k)}; \boldsymbol{\theta}^{(\tau_i^k)}, z_{i,m}^{(\tau_i^k)}) - \widehat{\mathcal{L}}_i'(\boldsymbol{\theta}^{(k)}, \boldsymbol{\theta} - \boldsymbol{\theta}^{(k)}; \boldsymbol{\theta}^{(\tau_i^k)}) \right\}}{\|\boldsymbol{\theta}^{(k)} - \boldsymbol{\theta}\|}.
$$

We have

$$
g^{(k)} \geq -\|\nabla \widehat{e}^{(k)}(\boldsymbol{\theta}^{(k)})\| + \inf_{\boldsymbol{\theta} \in \Theta} (-\epsilon^{(k)}(\boldsymbol{\theta})) \geq -\|\nabla \widehat{e}^{(k)}(\boldsymbol{\theta}^{(k)})\| - \sup_{\boldsymbol{\theta} \in \Theta} |\epsilon^{(k)}(\boldsymbol{\theta})| \,. \tag{25}
$$

Since $g^{(k)} = g_+^{(k)} - g_-^{(k)}$ and $g_+^{(k)} g_-^{(k)} = 0$, this implies

$$
g_-^{(k)} \leq \|\nabla \widehat{e}^{(k)}(\boldsymbol{\theta}^{(k)})\| + \sup_{\boldsymbol{\theta} \in \Theta} |\epsilon^{(k)}(\boldsymbol{\theta})| \,. \tag{26}
$$

Consider the above inequality when $k = K$, *i.e.,* the random index, and taking total expectations on both sides gives

$$
\mathbb{E}[g_-^{(K)}] \leq \mathbb{E}[\|\nabla \widehat{e}^{(K)}(\boldsymbol{\theta}^{(K)})\|] + \mathbb{E}[\sup_{\boldsymbol{\theta} \in \Theta} \epsilon^{(K)}(\boldsymbol{\theta})] \,.
$$

We note that

$$
\left( \mathbb{E}[\|\nabla \widehat{e}^{(K)}(\boldsymbol{\theta}^{(K)})\|] \right)^2 \leq \mathbb{E}[\|\nabla \widehat{e}^{(K)}(\boldsymbol{\theta}^{(K)})\|^2] \leq \frac{\Delta(K_{\max})}{K_{\max}} \,,
$$

where the first inequality is due to the convexity of $(\cdot)^2$ and the Jensen's inequality, and

$$
\begin{aligned}
\mathbb{E}[\sup_{\boldsymbol{\theta} \in \Theta} \epsilon^{(K)}(\boldsymbol{\theta})] &= \frac{1}{K_{\max}} \sum_{k=0}^{K_{\max}} \mathbb{E}[\sup_{\boldsymbol{\theta} \in \Theta} \epsilon^{(k)}(\boldsymbol{\theta})] \overset{(a)}{\leq} \frac{C_{\mathrm{gr}}}{K_{\max}} \sum_{k=0}^{K_{\max}-1} \mathbb{E}\left[ \frac{1}{n} \sum_{i=1}^n M_{(\tau_i^k)}^{-1/2} \right] \\
&\overset{(b)}{\leq} \frac{C_{\mathrm{gr}}}{K_{\max}} \sum_{k=0}^{K_{\max}-1} M_{(k)}^{-1/2} \,,
\end{aligned}
$$

where (a) is due to H4 and (b) is due to (24). This implies

$$
\mathbb{E}[g_-^{(K)}] \leq \sqrt{\frac{\Delta_{(K_{\max})}}{K_{\max}}} + \frac{C_{\mathrm{gr}}}{K_{\max}} \sum_{k=0}^{K_{\max}-1} M_{(k)}^{-1/2} \,,
$$

and concludes the proof of the theorem. $\qquad\square$

A.2 PROOF OF THEOREM 2

**Theorem.** *Under H1-H4. In addition, assume that $\{M_{(k)}\}_{k\geq 0}$ is a non-decreasing sequence of integers which satisfies $\sum_{k=0}^{\infty} M_{(k)}^{-1/2} < \infty$. Then:*

1. *the negative part of the stationarity measure converges a.s. to zero, i.e., $\lim_{k\to\infty} g_-(\boldsymbol{\theta}^{(k)}) \overset{a.s.}{=} 0$.*

2. *the objective value $\mathcal{L}(\boldsymbol{\theta}^{(k)})$ converges a.s. to a finite number $\underline{\mathcal{L}}$, i.e., $\lim_{k\to\infty} \mathcal{L}(\boldsymbol{\theta}^{(k)}) \overset{a.s.}{=} \underline{\mathcal{L}}$.*

**Proof** We apply the following auxiliary lemma which proof can be found in Appendix A.3 for the readability of the current proof:

**Lemma 1.** *Let $(V_k)_{k\geq 0}$ be a non negative sequence of random variables such that $\mathbb{E}[V_0] < \infty$. Let $(X_k)_{k\geq 0}$ a non negative sequence of random variables and $(E_k)_{k\geq 0}$ be a sequence of random variables such that $\sum_{k=0}^{\infty} \mathbb{E}[|E_k|] < \infty$. If for any $k \geq 1$:*

$$V_k \leq V_{k-1} - X_{k-1} + E_{k-1} \tag{27}$$

*then:*

*(i) for all $k \geq 0$, $\mathbb{E}[V_k] < \infty$ and the sequence $(V_k)_{k\geq 0}$ converges a.s. to a finite limit $V_\infty$.*

*(ii) the sequence $(\mathbb{E}[V_k])_{k\geq 0}$ converges and $\lim_{k\to\infty} \mathbb{E}[V_k] = \mathbb{E}[V_\infty]$.*

*(iii) the series $\sum_{k=0}^{\infty} X_k$ converges almost surely and $\sum_{k=0}^{\infty} \mathbb{E}[X_k] < \infty$.*

We proceed from (19) by re-arranging terms and observing that

$$\widehat{\mathcal{L}}^{(k+1)}(\boldsymbol{\theta}^{(k+1)}) \leq \widehat{\mathcal{L}}^{(k)}(\boldsymbol{\theta}^{(k)}) - \tfrac{1}{n}\big(\widehat{\mathcal{L}}_{i_k}(\boldsymbol{\theta}^{(k)};\boldsymbol{\theta}^{(\tau_{i_k}^k)}) - \widehat{\mathcal{L}}_{i_k}(\boldsymbol{\theta}^{(k)};\boldsymbol{\theta}^{(k)})\big)$$
$$- \big(\widetilde{\mathcal{L}}^{(k+1)}(\boldsymbol{\theta}^{(k+1)}) - \widehat{\mathcal{L}}^{(k+1)}(\boldsymbol{\theta}^{(k+1)})\big) + \big(\widetilde{\mathcal{L}}^{(k)}(\boldsymbol{\theta}^{(k)}) - \widehat{\mathcal{L}}^{(k)}(\boldsymbol{\theta}^{(k)})\big)$$
$$+ \tfrac{1}{n}\big(\widetilde{\mathcal{L}}_{i_k}(\boldsymbol{\theta}^{(k)};\boldsymbol{\theta}^{(k)}, \{z_{i_k,m}^{(k)}\}_{m=1}^{M_{(k)}}) - \widehat{\mathcal{L}}_{i_k}(\boldsymbol{\theta}^{(k)};\boldsymbol{\theta}^{(k)})\big)$$
$$+ \tfrac{1}{n}\big(\widehat{\mathcal{L}}_{i_k}(\boldsymbol{\theta}^{(k)};\boldsymbol{\theta}^{(\tau_{i_k}^k)}) - \widetilde{\mathcal{L}}_{i_k}(\boldsymbol{\theta}^{(k)};\boldsymbol{\theta}^{(\tau_{i_k}^k)}, \{z_{i_k,m}^{(\tau_{i_k}^k)}\}_{m=1}^{M_{(\tau_{i_k}^k)}})\big) \; .$$

Our idea is to apply Lemma 1. Under H1, the finite sum of surrogate functions $\widehat{\mathcal{L}}^{(k)}(\boldsymbol{\theta})$, defined in (15), is lower bounded by a constant $c_k > -\infty$ for any $\boldsymbol{\theta}$. To this end, we observe that

$$V_k := \widehat{\mathcal{L}}^{(k)}(\boldsymbol{\theta}^{(k)}) - \inf_{k\geq 0} c_k \geq 0 \tag{28}$$

is a non-negative random variable.

Secondly, under H1, the following random variable is non-negative

$$X_k := \tfrac{1}{n}\big(\widehat{\mathcal{L}}_{i_k}(\boldsymbol{\theta}^{(\tau_{i_k}^k)};\boldsymbol{\theta}^{(k)}) - \widehat{\mathcal{L}}_{i_k}(\boldsymbol{\theta}^{(k)};\boldsymbol{\theta}^{(k)})\big) \geq 0 \; . \tag{29}$$

Thirdly, we define

$$E_k = -\big(\widetilde{\mathcal{L}}^{(k+1)}(\boldsymbol{\theta}^{(k+1)}) - \widehat{\mathcal{L}}^{(k+1)}(\boldsymbol{\theta}^{(k+1)})\big) + \big(\widetilde{\mathcal{L}}^{(k)}(\boldsymbol{\theta}^{(k)}) - \widehat{\mathcal{L}}^{(k)}(\boldsymbol{\theta}^{(k)})\big)$$
$$+ \tfrac{1}{n}\big(\widetilde{\mathcal{L}}_{i_k}(\boldsymbol{\theta}^{(k)};\boldsymbol{\theta}^{(k)}, \{z_{i_k,m}^{(k)}\}_{m=1}^{M_{(k)}}) - \widehat{\mathcal{L}}_{i_k}(\boldsymbol{\theta}^{(k)};\boldsymbol{\theta}^{(k)})\big) \tag{30}$$
$$+ \tfrac{1}{n}\big(\widehat{\mathcal{L}}_{i_k}(\boldsymbol{\theta}^{(k)};\boldsymbol{\theta}^{(\tau_{i_k}^k)}) - \widetilde{\mathcal{L}}_{i_k}(\boldsymbol{\theta}^{(k)};\boldsymbol{\theta}^{(\tau_{i_k}^k)}, \{z_{i_k,m}^{(\tau_{i_k}^k)}\}_{m=1}^{M_{(\tau_{i_k}^k)}})\big) \; .$$

Note that from the definitions (28), (29), (30), we have $V_{k+1} \leq V_k - X_k + E_k$ for any $k \geq 1$.

Under H4, we observe that

$$\mathbb{E}\big[|\widetilde{\mathcal{L}}_{i_k}(\boldsymbol{\theta}^{(k)};\boldsymbol{\theta}^{(k)}, \{z_{i_k,m}^{(k)}\}_{m=1}^{M_{(k)}}) - \widehat{\mathcal{L}}_{i_k}(\boldsymbol{\theta}^{(k)};\boldsymbol{\theta}^{(k)})|\big] \leq C_r M_{(k)}^{-1/2}$$

$$\mathbb{E}\Big[\big|\widehat{\mathcal{L}}_{i_k}(\boldsymbol{\theta}^{(k)};\boldsymbol{\theta}^{(\tau_{i_k}^k)}) - \widetilde{\mathcal{L}}_{i_k}(\boldsymbol{\theta}^{(k)};\boldsymbol{\theta}^{(\tau_{i_k}^k)}, \{z_{i_k,m}^{(\tau_{i_k}^k)}\}_{m=1}^{M_{(\tau_{i_k}^k)}})\big|\Big] \leq C_r \mathbb{E}\Big[M_{(\tau_{i_k}^k)}^{-1/2}\Big]$$

$$\mathbb{E}\big[|\widetilde{\mathcal{L}}^{(k)}(\boldsymbol{\theta}^{(k)}) - \widehat{\mathcal{L}}^{(k)}(\boldsymbol{\theta}^{(k)})|\big] \leq \tfrac{1}{n}\sum_{i=1}^{n}C_{\mathsf{r}}\mathbb{E}\Big[M_{(\tau_i^k)}^{-1/2}\Big]$$

Therefore,

$$\mathbb{E}\big[|E_k|\big] \leq \tfrac{C_{\mathsf{r}}}{n}\Big(M_{(k)}^{-1/2} + \mathbb{E}\Big[M_{(\tau_{i_k}^k)}^{-1/2} + \sum_{i=1}^{n}\big\{M_{(\tau_i^k)}^{-1/2} + M_{(\tau_i^{k+1})}^{-1/2}\big\}\Big]\Big) \ .$$

Using (24) and the assumption on the sequence $\{M_{(k)}\}_{k\geq 0}$, we obtain that

$$\sum_{k=0}^{\infty}\mathbb{E}\big[|E_k|\big] < \frac{C_{\mathsf{r}}}{n}(2+2n)\sum_{k=0}^{\infty}M_{(k)}^{-1/2} < \infty.$$

Therefore, the conclusions in Lemma 1 hold. Precisely, we have $\sum_{k=0}^{\infty}X_k < \infty$ and $\sum_{k=0}^{\infty}\mathbb{E}[X_k] < \infty$ almost surely. Note that this implies

$$\infty > \sum_{k=0}^{\infty}\mathbb{E}[X_k] = \frac{1}{n}\sum_{k=0}^{\infty}\mathbb{E}\big[\widehat{\mathcal{L}}_{i_k}(\boldsymbol{\theta}^{(k)}; \boldsymbol{\theta}^{(\tau_{i_k}^k)}) - \widehat{\mathcal{L}}_{i_k}(\boldsymbol{\theta}^{(k)}; \boldsymbol{\theta}^{(k)})\big]$$

$$= \frac{1}{n}\sum_{k=0}^{\infty}\mathbb{E}\big[\widehat{\mathcal{L}}^{(k)}(\boldsymbol{\theta}^{(k)}) - \mathcal{L}(\boldsymbol{\theta}^{(k)})\big] = \frac{1}{n}\sum_{k=0}^{\infty}\mathbb{E}\big[\widehat{e}^{(k)}(\boldsymbol{\theta}^{(k)})\big] \ .$$

Since $\widehat{e}^{(k)}(\boldsymbol{\theta}^{(k)}) \geq 0$, the above implies

$$\lim_{k\to\infty}\widehat{e}^{(k)}(\boldsymbol{\theta}^{(k)}) = 0 \quad \text{a.s.} \tag{31}$$

and subsequently applying (18), we have $\lim_{k\to\infty}\|\widehat{e}^{(k)}(\boldsymbol{\theta}^{(k)})\| = 0$ almost surely. Finally, it follows from (18) and (26) that

$$\lim_{k\to\infty}g_-^{(k)} \leq \lim_{k\to\infty}\sqrt{2L}\sqrt{\widehat{e}^{(k)}(\boldsymbol{\theta}^{(k)})} + \lim_{k\to\infty}\sup_{\boldsymbol{\theta}\in\Theta}|\epsilon^{(k)}(\boldsymbol{\theta})| = 0 \ , \tag{32}$$

where the last equality holds almost surely due to the fact that $\sum_{k=0}^{\infty}\mathbb{E}[\sup_{\boldsymbol{\theta}\in\Theta}|\epsilon^{(k)}(\boldsymbol{\theta})|] < \infty$. This concludes the asymptotic convergence of the MISSO method.

Finally, we prove that $\mathcal{L}(\boldsymbol{\theta}^{(k)})$ converges almost surely. As a consequence of Lemma 1, it is clear that $\{V_k\}_{k\geq 0}$ converges almost surely and so is $\{\widehat{\mathcal{L}}^{(k)}(\boldsymbol{\theta}^{(k)})\}_{k\geq 0}$, *i.e.*, we have $\lim_{k\to\infty}\widehat{\mathcal{L}}^{(k)}(\boldsymbol{\theta}^{(k)}) = \underline{\mathcal{L}}$. Applying (31) implies that

$$\underline{\mathcal{L}} = \lim_{k\to\infty}\widehat{\mathcal{L}}^{(k)}(\boldsymbol{\theta}^{(k)}) = \lim_{k\to\infty}\mathcal{L}(\boldsymbol{\theta}^{(k)}) \quad \text{a.s.}$$

This shows that $\mathcal{L}(\boldsymbol{\theta}^{(k)})$ converges almost surely to $\underline{\mathcal{L}}$. $\qquad\square$

## A.3 PROOF OF LEMMA 1

**Lemma.** *Let* $(V_k)_{k\geq 0}$ *be a non negative sequence of random variables such that* $\mathbb{E}[V_0] < \infty$. *Let* $(X_k)_{k\geq 0}$ *a non negative sequence of random variables and* $(E_k)_{k\geq 0}$ *be a sequence of random variables such that* $\sum_{k=0}^{\infty}\mathbb{E}[|E_k|] < \infty$. *If for any* $k \geq 1$:

$$V_k \leq V_{k-1} - X_{k-1} + E_{k-1}$$

*then:*

    *(i) for all* $k \geq 0$, $\mathbb{E}[V_k] < \infty$ *and the sequence* $(V_k)_{k\geq 0}$ *converges a.s. to a finite limit* $V_\infty$.

    *(ii) the sequence* $(\mathbb{E}[V_k])_{k\geq 0}$ *converges and* $\lim_{k\to\infty}\mathbb{E}[V_k] = \mathbb{E}[V_\infty]$.

    *(iii) the series* $\sum_{k=0}^{\infty}X_k$ *converges almost surely and* $\sum_{k=0}^{\infty}\mathbb{E}[X_k] < \infty$.

**Proof** We first show that for all $k \geq 0$, $\mathbb{E}[V_k] < \infty$. Note indeed that:

$$0 \leq V_k \leq V_0 - \sum_{j=1}^{k} X_j + \sum_{j=1}^{k} E_j \leq V_0 + \sum_{j=1}^{k} E_j \ , \tag{33}$$

showing that $\mathbb{E}[V_k] \leq \mathbb{E}[V_0] + \mathbb{E}\left[\sum_{j=1}^{k} E_j\right] < \infty$.

Since $0 \leq X_k \leq V_{k-1} - V_k + E_k$ we also obtain for all $k \geq 0$, $\mathbb{E}[X_k] < \infty$. Moreover, since $\mathbb{E}\left[\sum_{j=1}^{\infty} |E_j|\right] < \infty$, the series $\sum_{j=1}^{\infty} E_j$ converges a.s. We may therefore define:

$$W_k = V_k + \sum_{j=k+1}^{\infty} E_j \tag{34}$$

Note that $\mathbb{E}[|W_k|] \leq \mathbb{E}[V_k] + \mathbb{E}\left[\sum_{j=k+1}^{\infty} |E_j|\right] < \infty$. For all $k \geq 1$, we get:

$$W_k \leq V_{k-1} - X_k + \sum_{j=k}^{\infty} E_j \leq W_{k-1} - X_k \leq W_{k-1} \tag{35}$$

$$\mathbb{E}[W_k] \leq \mathbb{E}[W_{k-1}] - \mathbb{E}[X_k] \ .$$

Hence the sequences $(W_k)_{k \geq 0}$ and $(\mathbb{E}[W_k])_{k \geq 0}$ are non increasing. Since for all $k \geq 0$, $W_k \geq -\sum_{j=1}^{\infty} |E_j| > -\infty$ and $\mathbb{E}[W_k] \geq -\sum_{j=1}^{\infty} \mathbb{E}[|E_j|] > -\infty$, the (random) sequence $(W_k)_{k \geq 0}$ converges a.s. to a limit $W_\infty$ and the (deterministic) sequence $(\mathbb{E}[W_k])_{k \geq 0}$ converges to a limit $w_\infty$. Since $|W_k| \leq V_0 + \sum_{j=1}^{\infty} |E_j|$, the Fatou lemma implies that:

$$\mathbb{E}[\liminf_{k \to \infty} |W_k|] = \mathbb{E}[|W_\infty|] \leq \liminf_{k \to \infty} \mathbb{E}[|W_k|] \leq \mathbb{E}[V_0] + \sum_{j=1}^{\infty} \mathbb{E}[|E_j|] < \infty \ , \tag{36}$$

showing that the random variable $W_\infty$ is integrable.

In the sequel, set $U_k \triangleq W_0 - W_k$. By construction we have for all $k \geq 0$, $U_k \geq 0$, $U_k \leq U_{k+1}$ and $\mathbb{E}[U_k] \leq \mathbb{E}[|W_0|] + \mathbb{E}[|W_k|] < \infty$ and by the monotone convergence theorem, we get:

$$\lim_{k \to \infty} \mathbb{E}[U_k] = \mathbb{E}[\lim_{k \to \infty} U_k] \ . \tag{37}$$

Finally, we have:

$$\lim_{k \to \infty} \mathbb{E}[U_k] = \mathbb{E}[W_0] - w_\infty \quad \text{and} \quad \mathbb{E}[\lim_{k \to \infty} U_k] = \mathbb{E}[W_0] - \mathbb{E}[W_\infty] \ . \tag{38}$$

showing that $\mathbb{E}[W_\infty] = w_\infty$ and concluding the proof of (ii). Moreover, using (35) we have that $W_k \leq W_{k-1} - X_k$ which yields:

$$\sum_{j=1}^{\infty} X_j \leq W_0 - W_\infty < \infty \ ,$$

$$\sum_{j=1}^{\infty} \mathbb{E}[X_j] \leq \mathbb{E}[W_0] - w_\infty < \infty \ , \tag{39}$$

an concludes the proof of the lemma. $\square$

# B PRACTICAL DETAILS FOR THE BINARY LOGISTIC REGRESSION ON THE TRAUMABASE

## B.1 TRAUMABASE DATASET QUANTITATIVE VARIABLES

The list of the 16 quantitative variables we use in our experiments are as follows — *age, weight, height, BMI (Body Mass Index), the Glasgow Coma Scale, the Glasgow Coma Scale motor component, the minimum systolic blood pressure, the minimum diastolic blood pressure, the maximum*

*number of heart rate (or pulse) per unit time (usually a minute), the systolic blood pressure at arrival of ambulance, the diastolic blood pressure at arrival of ambulance, the heart rate at arrival of ambulance, the capillary Hemoglobin concentration, the oxygen saturation, the fluid expansion colloids, the fluid expansion cristalloids, the pulse pressure for the minimum value of diastolic and systolic blood pressure, the pulse pressure at arrival of ambulance.*

### B.2 METROPOLIS-HASTINGS ALGORITHM

During the simulation step of the MISSO method, the sampling from the target distribution $\pi(z_{i,\mathrm{mis}}; \boldsymbol{\theta}) := p(z_{i,\mathrm{mis}}|z_{i,\mathrm{obs}}, y_i; \boldsymbol{\theta})$ is performed using a Metropolis-Hastings (MH) algorithm (Meyn & Tweedie, 2012) with proposal distribution $q(z_{i,\mathrm{mis}}; \boldsymbol{\delta}) := p(z_{i,\mathrm{mis}}|z_{i,\mathrm{obs}}; \boldsymbol{\delta})$ where $\boldsymbol{\theta} = (\beta, \Omega)$ and $\boldsymbol{\delta} = (\xi, \Sigma)$. The parameters of the Gaussian conditional distribution of $z_{i,\mathrm{mis}}|z_{i,\mathrm{obs}}$ read:

$$\xi = \beta_{miss} + \Omega_{mis,obs}\Omega_{obs,obs}^{-1}(z_{i,\mathrm{obs}} - \beta_{obs}) ,$$

$$\Sigma = \Omega_{mis,mis} + \Omega_{mis,obs}\Omega_{obs,obs}^{-1}\Omega_{obs,\mathrm{mis}} ,$$

where we have used the Schur Complement of $\Omega_{obs,obs}$ in $\Omega$ and noted $\beta_{mis}$ (resp. $\beta_{obs}$) the missing (resp. observed) elements of $\beta$. The MH algorithm is summarized in Algorithm 3.

---

**Algorithm 3** MH aglorithm

1: **Input:** initialization $z_{i,\mathrm{mis},0} \sim q(z_{i,\mathrm{mis}}; \boldsymbol{\delta})$
2: **for** $m = 1, \cdots, M$ **do**
3:     Sample $z_{i,\mathrm{mis},m} \sim q(z_{i,\mathrm{mis}}; \boldsymbol{\delta})$
4:     Sample $u \sim \mathcal{U}(\llbracket 0, 1 \rrbracket)$
5:     Calculate the ratio $r = \frac{\pi(z_{i,\mathrm{mis},m};\boldsymbol{\theta})/q(z_{i,\mathrm{mis},m});\boldsymbol{\delta})}{\pi(z_{i,\mathrm{mis},m-1};\boldsymbol{\theta})/q(z_{i,\mathrm{mis},m-1);\boldsymbol{\delta})}$
6:     **if** $u < r$ **then**
7:         Accept $z_{i,\mathrm{mis},m}$
8:     **else**
9:         $z_{i,\mathrm{mis},m} \leftarrow z_{i,\mathrm{mis},m-1}$
10:    **end if**
11: **end for**
12: **Output:** $z_{i,\mathrm{mis},M}$

---

### B.3 MISSO UPDATE

**Choice of surrogate function for MISO:** We recall the MISO deterministic surrogate defined in (7):

$$\widehat{\mathcal{L}}_i(\boldsymbol{\theta}; \overline{\boldsymbol{\theta}}) = \int_{\mathsf{Z}} \log \left( p_i(z_{i,\mathrm{mis}}, \overline{\boldsymbol{\theta}})/f_i(z_{i,\mathrm{mis}}, \boldsymbol{\theta}) \right) p_i(z_{i,\mathrm{mis}}, \overline{\boldsymbol{\theta}})\mu_i(\mathrm{d}z_i) .$$

where $\boldsymbol{\theta} = (\delta, \beta, \Omega)$ and $\overline{\boldsymbol{\theta}} = (\overline{\delta}, \overline{\beta}, \overline{\Omega})$. We adapt it to our missing covariates problem and decompose the surrogate function defined above into an observed and a missing part.

**Surrogate function decomposition** We adapt it to our missing covariates problem and decompose the term depending on $\theta$, while $\bar{\theta}$ is fixed, in two following parts leading to

$$\widehat{\mathcal{L}}_i(\boldsymbol{\theta}; \overline{\boldsymbol{\theta}})$$

$$= - \int_{\mathsf{Z}} \log f_i(z_{i,\mathrm{mis}}, z_{i,\mathrm{obs}}, \boldsymbol{\theta}) p_i(z_{i,\mathrm{mis}}, \overline{\boldsymbol{\theta}})\mu_i(\mathrm{d}z_{i,\mathrm{mis}})$$

$$= - \int_{\mathsf{Z}} \log \left[ p_i(y_i|z_{i,\mathrm{mis}}, z_{i,\mathrm{obs}}, \delta) p_i(z_{i,\mathrm{mis}}, \beta, \Omega) \right] p_i(z_i, \overline{\boldsymbol{\theta}})\mu_i(\mathrm{d}z_{i,\mathrm{mis}})$$

$$= \underbrace{- \int_{\mathsf{Z}} \log p_i(y_i|z_{i,\mathrm{mis}}, z_{i,\mathrm{obs}}, \delta) p_i(z_i, \overline{\boldsymbol{\theta}})\mu_i(\mathrm{d}z_{i,\mathrm{mis}})}_{=\hat{\mathcal{L}}_i^{(1)}(\delta, \overline{\boldsymbol{\theta}})} \underbrace{- \int_{\mathsf{Z}} \log p_i(z_{i,\mathrm{mis}}, \beta, \Omega) p_i(z_i, \overline{\boldsymbol{\theta}})\mu_i(\mathrm{d}z_{i,\mathrm{mis}})}_{=\hat{\mathcal{L}}_i^{(2)}(\beta, \Omega, \overline{\boldsymbol{\theta}})} .$$

$$(40)$$

The mean $\beta$ and the covariance $\Omega$ of the latent structure can be estimated minimizing the sum of MISSO surrogates $\tilde{\mathcal{L}}_i^{(2)}(\beta, \Omega, \overline{\boldsymbol{\theta}}, \{z_m\}_{m=1}^M)$, defined as MC approximation of $\hat{\mathcal{L}}_i^{(2)}(\beta, \Omega, \overline{\boldsymbol{\theta}})$, for all $i \in [\![n]\!]$, in closed-form expression.

We thus keep the surrogate $\hat{\mathcal{L}}_i^{(2)}(\beta, \Omega, \overline{\boldsymbol{\theta}})$ as it is, and consider the following quadratic approximation of $\hat{\mathcal{L}}_i^{(1)}(\delta, \overline{\boldsymbol{\theta}})$ to estimate the vector of logistic parameters $\delta$:

$$\hat{\mathcal{L}}_i^{(1)}(\delta, \overline{\boldsymbol{\theta}}) - \int_{\mathsf{Z}} \nabla \log p_i(y_i|z_{i,\text{mis}}, z_{i,\text{obs}}, \delta)\big|_{\delta=\bar{\delta}} \, p_i(z_{i,\text{mis}}, \overline{\boldsymbol{\theta}}) \mu_i(dz_{i,\text{mis}})(\delta - \bar{\delta})$$
$$- (\delta - \bar{\delta})/2 \int_{\mathsf{Z}} \nabla^2 \log p_i(y_i|z_{i,\text{mis}}, z_{i,\text{obs}}, \delta) p_i(z_{i,\text{mis}}, \overline{\boldsymbol{\theta}}) p_i(z_{i,\text{mis}}, \overline{\boldsymbol{\theta}}) \mu_i(dz_{i,\text{mis}})(\delta - \bar{\delta})^\top.$$

Recall that:

$$\nabla \log p_i(y_i|z_{i,\text{mis}}, z_{i,\text{obs}}, \delta) = z_i \left( y_i - S(\delta^\top z_i) \right) \ ,$$
$$\nabla^2 \log p_i(y_i|z_{i,\text{mis}}, z_{i,\text{obs}}, \delta) = -z_i z_i^\top \dot{S}(\delta^\top z_i) \ ,$$

where $\dot{S}(u)$ is the derivative of $S(u)$. Note that $\dot{S}(u) \le 1/4$ and since, for all $i \in [\![n]\!]$, the $p \times p$ matrix $z_i z_i^\top$ is semi-definite positive we can assume that:

**L1.** *For all $i \in [\![n]\!]$ and $\epsilon > 0$, there exist, for all $z_i \in \mathsf{Z}$, a positive definite matrix $H_i(z_i) := \frac{1}{4}(z_i z_i^\top + \epsilon I_d)$ such that for all $\delta \in \mathbb{R}^p$, $-z_i z_i^\top \dot{S}(\delta^\top z_i) \le H_i(z_i)$.*

Then, we use, for all $i \in [\![n]\!]$, the following surrogate function to estimate $\delta$:

$$\bar{\mathcal{L}}_i^{(1)}(\delta, \overline{\boldsymbol{\theta}}) = \hat{\mathcal{L}}_i^{(1)}(\bar{\delta}, \overline{\boldsymbol{\theta}}) - D_i^\top (\delta - \bar{\delta}) + \frac{1}{2}(\delta - \bar{\delta}) H_i (\delta - \bar{\delta})^\top \ , \tag{41}$$

where:

$$D_i = \int_{\mathsf{Z}} \nabla \log p_i(y_i|z_{i,\text{mis}}, z_{i,\text{obs}}, \delta)\big|_{\delta=\bar{\delta}} \, p_i(z_{i,\text{mis}}, \overline{\boldsymbol{\theta}}) \mu_i(dz_{i,\text{mis}}) \ ,$$
$$H_i = \int_{\mathsf{Z}} H_i(z_{i,\text{mis}}) p_i(z_{i,\text{mis}}, \overline{\boldsymbol{\theta}}) \mu_i(dz_{i,\text{mis}}) \ .$$

Finally, at iteration $k$, the total surrogate is:

$$\tilde{\mathcal{L}}^{(k)}(\theta) = \frac{1}{n} \sum_{i=1}^n \tilde{\mathcal{L}}_i(\theta, \theta^{(\tau_i^k)}, \{z_{i,m}\}_{m=1}^{M_{(\tau_i^k)}})$$
$$= \frac{1}{n} \sum_{i=1}^n \tilde{\mathcal{L}}_i^{(2)}(\beta, \Omega, \theta^{(\tau_i^k)}, \{z_{i,m}\}_{m=1}^{M_{(\tau_i^k)}}) - \frac{1}{n} \sum_{i=1}^n \tilde{D}_i^{(\tau_i^k)}(\delta - \delta^{(\tau_i^k)}) \tag{42}$$
$$+ \frac{1}{2n} \sum_{i=1}^n (\delta - \delta^{(\tau_i^k)}) \left\{ \tilde{H}_i^{(\tau_i^k)} \right\} (\delta - \delta^{(\tau_i^k)})^\top \ ,$$

where for all $i \in [\![n]\!]$:

$$\tilde{D}_i^{(\tau_i^k)} = \frac{1}{M_{(\tau_i^k)}} \sum_{m=1}^{M_{(\tau_i^k)}} z_{i,m}^{(\tau_i^k)} \left( y_i - S((\delta^{(\tau_i^k)})^\top z_{i,m}(\tau_i^k)) \right) \ ,$$
$$\tilde{H}_i^{(\tau_i^k)} = \frac{1}{4M_{(\tau_i^k)}} \sum_{m=1}^{M_{(\tau_i^k)}} z_{i,m}^{(\tau_i^k)} (z_{i,m}^{(\tau_i^k)})^\top \ .$$

Minimizing the total surrogate (42) boils down to performing a quasi-Newton step. It is perhaps sensible to apply some diagonal loading which is perfectly compatible with the surrogate interpretation we just gave.

The logistic parameters are estimated as follows:

$$\boldsymbol{\delta}^{(k)} = \arg \min_{\delta \in \Theta} \frac{1}{n} \sum_{i=1}^n \tilde{\mathcal{L}}_i^{(1)}(\delta, \theta^{(\tau_i^k)}, \{z_{i,m}\}_{m=1}^{M_{(\tau_i^k)}}) \ ,$$

where $\tilde{\mathcal{L}}_i^{(1)}(\delta, \theta^{(\tau_i^k)}, \{z_{i,m}\}_{m=1}^{M_{(\tau_i^k)}})$ is the MC approximation of the MISO surrogate defined in (41) and which leads to the following quasi-Newton step:

$$\boldsymbol{\delta}^{(k)} = \frac{1}{n}\sum_{i=1}^{n}\boldsymbol{\delta}^{(\tau_i^k)} - (\tilde{H}^{(k)})^{-1}\tilde{D}^{(k)} \,,$$

with $\tilde{D}^{(k)} = \frac{1}{n}\sum_{i=1}^{n}\tilde{D}_i^{(\tau_i^k)}$ and $\tilde{H}^{(k)} = \frac{1}{n}\sum_{i=1}^{n}\tilde{H}_i^{(\tau_i^k)}$.

**MISSO updates:** At the $k$-th iteration, and after the initialization, for all $i \in [\![n]\!]$, of the latent variables $(z_i^{(0)})$, the MISSO algorithm consists in picking an index $i_k$ uniformly on $[\![n]\!]$, completing the observations by sampling a Monte Carlo batch $\{z_{i_k,\mathrm{mis},m}^{(k)}\}_{m=1}^{M_{(k)}}$ of missing values from the conditional distribution $p(z_{i_k,\mathrm{mis}}|z_{i_k,\mathrm{obs}}, y_{i_k}; \boldsymbol{\theta}^{(k-1)})$ using an MCMC sampler and computing the estimated parameters as follows:

$$\boldsymbol{\beta}^{(k)} = \arg\min_{\beta\in\Theta}\frac{1}{n}\sum_{i=1}^{n}\tilde{\mathcal{L}}_i^{(2)}(\beta, \Omega^{(k)}, \theta^{(\tau_i^k)}, \{z_{i,m}\}_{m=1}^{M_{(\tau_i^k)}}) = \frac{1}{n}\sum_{i=1}^{n}\frac{1}{M_{(\tau_i^k)}}\sum_{m=1}^{M_{(\tau_i^k)}}z_{i,m}^{(k)} \,,$$

$$\boldsymbol{\Omega}^{(k)} = \arg\min_{\Omega\in\Theta}\frac{1}{n}\sum_{i=1}^{n}\tilde{\mathcal{L}}_i^{(2)}(\beta^{(k)}, \Omega, \theta^{(\tau_i^k)}, \{z_{i,m}\}_{m=1}^{M_{(\tau_i^k)}}) = \frac{1}{n}\sum_{i=1}^{n}\frac{1}{M_{(\tau_i^k)}}\sum_{m=1}^{M_{(\tau_i^k)}}w_{i,m}^{(k)} \,, \qquad (43)$$

$$\boldsymbol{\delta}^{(k)} = \frac{1}{n}\sum_{i=1}^{n}\boldsymbol{\delta}^{(\tau_i^k)} - (\tilde{H}^{(k)})^{-1}\tilde{D}^{(k)} \,.$$

where $z_{i,m}^{(k)} = (z_{i,\mathrm{mis},m}^{(k)}, z_{i,\mathrm{obs}})$ is composed of a simulated and an observed part, $\tilde{D}^{(k)} = \frac{1}{n}\sum_{i=1}^{n}\tilde{D}_i^{(\tau_i^k)}$, $\tilde{H}^{(k)} = \frac{1}{n}\sum_{i=1}^{n}\tilde{H}_i^{(\tau_i^k)}$ and $w_{i,m}^{(k)} = z_{i,m}^{(k)}(z_{i,m}^{(k)})^\top - \beta^{(k)}(\beta^{(k)})^\top$. Besides, $\tilde{\mathcal{L}}_i^{(1)}(\beta, \Omega, \overline{\theta}, \{z_m\}_{m=1}^M)$ and $\tilde{\mathcal{L}}_i^{(2)}(\beta, \Omega, \overline{\theta}, \{z_m\}_{m=1}^M)$ are defined as MC approximation of $\hat{\mathcal{L}}_i^{(1)}(\beta, \Omega, \overline{\theta})$ and $\hat{\mathcal{L}}_i^{(2)}(\beta, \Omega, \overline{\theta})$, for all $i \in [\![n]\!]$ as components of the surrogate function (40).

## B.4 WALL CLOCK TIME

We provide Table 1, the running time for each method, plotted in Figure 1, employed to train a logistic regression with missing values on the TraumaBase dataset ($p = 16$ influential quantitative measurements, on $n = 6384$ patients).

The running times are sensibly the same since for each method the computation complexity per epoch is similar. We remark a slight delay using the MISSO method with a batch size of 1, as our code implemented in R, is not totally optimized and parallelized. Yet, when the batch size tends to 100%, we retrieve the duration of MCEM, which is consistent with the fact that MISSO with a full batch update boils down to the MCEM algorithm.

Table 1: Logistic Regression with missing values: running time in seconds for 10 epochs.

|  | SAEM | MCEM | MISSO | MISSO10 | MISSO50 |
|---|---|---|---|---|---|
| Logistic Regression | 2033.2 | 1972.4 | 2244.8 | 2139.4 | 2005.2 |

We plot Figure 3, the updated parameters for the Logistic regression example against the time elapsed (in seconds).

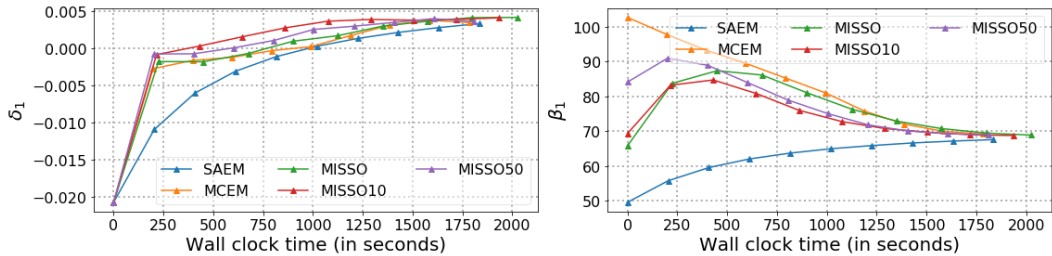

Figure 3: Convergence of parameters $\delta$ and $\beta$ for the SAEM, the MCEM and the MISSO methods. The convergence is plotted against time elapsed (in seconds).

## C    PRACTICAL DETAILS FOR THE INCREMENTAL VARIATIONAL INFERENCE

### C.1    NEURAL NETWORKS ARCHITECTURE

**Bayesian LeNet-5 Architecture:** We describe in Table 2 the architecture of the Convolutional Neural Network introduced in (LeCun et al., 1998) and trained on MNIST:

| layer type | width | stride | padding | input shape | nonlinearity |
|---|---|---|---|---|---|
| convolution ($5 \times 5$) | 6 | 1 | 0 | $1 \times 32 \times 32$ | ReLU |
| max-pooling ($2 \times 2$) |  | 2 | 0 | $6 \times 28 \times 28$ |  |
| convolution ($5 \times 5$) | 6 | 1 | 0 | $1 \times 14 \times 14$ | ReLU |
| max-pooling ($2 \times 2$) |  | 2 | 0 | $16 \times 10 \times 10$ |  |
| fully-connected | 120 |  |  | 400 | ReLU |
| fully-connected | 84 |  |  | 120 | ReLU |
| fully-connected | 10 |  |  | 84 |  |

Table 2: LeNet-5 architecture

**Bayesian ResNet-18 Architecture:** We describe in Table 3 the architecture of the Resnet-18 we train on CIFAR-10:

| layer type | Output Size | ResNet-18 | nonlinearity |
|---|---|---|---|
| conv1 | $112 \times 112 \times 64$ | $7 \times 7$, 64, stride 2 | ReLU |
| conv2x | $56 \times 56 \times 64$ | $\begin{pmatrix} 3 \times 3, 64 \\ 3 \times 3, 64 \end{pmatrix} \times 2$ | ReLU |
| conv3x | $28 \times 28 \times 128$ | $\begin{pmatrix} 3 \times 3, 128 \\ 3 \times 3, 128 \end{pmatrix} \times 2$ | ReLU |
| conv4x | $14 \times 14 \times 256$ | $\begin{pmatrix} 3 \times 3, 256 \\ 3 \times 3, 256 \end{pmatrix} \times 2$ | ReLU |
| conv5x | $7 \times 7 \times 512$ | $\begin{pmatrix} 3 \times 3, 512 \\ 3 \times 3, 512 \end{pmatrix} \times 2$ | ReLU |
| average pool | $1 \times 1 \times 512$ | $7 \times 7$ average pool | ReLU |
| fully connected | 1000 | $512 \times 1000$ fully connections |  |
| softmax | 1000 |  |  |

Table 3: ResNet-18 architecture

### C.2    ALGORITHMS UPDATES

First, we initialize the means $\mu_\ell^{(0)}$ for $\ell \in [\![d]\!]$ and variance estimates $\sigma^{(0)}$. At iteration $k$, minimizing the sum of stochastic surrogates defined as in (6) and (13) yields the following MISSO update —

step (i) pick a function index $i_k$ uniformly on $[\![n]\!]$; step (ii) sample a Monte Carlo batch $\{z_m^{(k)}\}_{m=1}^{M_{(k)}}$ from $\mathcal{N}(0, \mathbf{I})$; and step (iii) update the parameters as

$$\mu_\ell^{(k)} = \frac{1}{n}\sum_{i=1}^{n}\mu_\ell^{(\tau_i^k)} - \frac{\gamma}{n}\sum_{i=1}^{n}\hat{\boldsymbol{\delta}}_{\mu_\ell,i}^{(k)} \quad \text{and} \quad \sigma^{(k)} = \frac{1}{n}\sum_{i=1}^{n}\sigma^{(\tau_i^k)} - \frac{\gamma}{n}\sum_{i=1}^{n}\hat{\boldsymbol{\delta}}_{\sigma,i}^{(k)}, \tag{44}$$

where we define the following gradient terms for all $i \in [\![1, n]\!]$:

$$\hat{\boldsymbol{\delta}}_{\mu_\ell,i}^{(k)} = -\frac{1}{M_{(k)}}\sum_{m=1}^{M_{(k)}}\nabla_w \log p(y_i|x_i, w)\Big|_{w=t(\boldsymbol{\theta}^{(k-1)}, z_m^{(k)})} + \nabla_{\mu_\ell}d(\boldsymbol{\theta}^{(k-1)}),$$

$$\hat{\boldsymbol{\delta}}_{\sigma,i}^{(k)} = -\frac{1}{M_{(k)}}\sum_{m=1}^{M_{(k)}}z_m^{(k)}\nabla_w \log p(y_i|x_i, w)\Big|_{w=t(\boldsymbol{\theta}^{(k-1)}, z_m^{(k)})} + \nabla_\sigma d(\boldsymbol{\theta}^{(k-1)}). \tag{45}$$

Note that our analysis in the main text does require the parameter to be in a compact set. For the current estimation problem considered, this can be enforced in practice by restricting the parameters in a ball. In our simulation for the BNNs example, we did not implement the algorithms that stick closely to the compactness requirement for illustrative purposes. However, we observe empirically that the parameters are always bounded. The update rules can be easily modified to respect the requirement. For the considered VI problem, we recall the surrogate functions (11) are quadratic and indeed a simple projection step suffices to ensure boundedness of the iterates.

For all benchmark algorithms, we pick, at iteration $k$, a function index $i_k$ uniformly on $[\![n]\!]$ and sample a Monte Carlo batch $\{z_m^{(k)}\}_{m=1}^{M_{(k)}}$ from the standard Gaussian distribution. The updates of the parameters $\mu_\ell$ for all $\ell \in [\![d]\!]$ and $\sigma$ break down as follows:

**Monte Carlo SAG update**: Set

$$\mu_\ell^{(k)} = \mu_\ell^{(k-1)} - \frac{\gamma}{n}\sum_{i=1}^{n}\hat{\boldsymbol{\delta}}_{\mu_\ell,i}^{(k)} \quad \text{and} \quad \sigma^{(k)} = \sigma^{(k-1)} - \frac{\gamma}{n}\sum_{i=1}^{n}\hat{\boldsymbol{\delta}}_{\sigma,i}^{(k)},$$

where $\hat{\boldsymbol{\delta}}_{\mu_\ell,i}^{(k)} = \hat{\boldsymbol{\delta}}_{\mu_\ell,i}^{(k-1)}$ and $\hat{\boldsymbol{\delta}}_{\sigma,i}^{(k)} = \hat{\boldsymbol{\delta}}_{\sigma,i}^{(k-1)}$ for $i \neq i_k$ and are defined by (45) for $i = i_k$. The learning rate is set to $\gamma = 10^{-3}$.

**Bayes By Backprop update**: Set

$$\mu_\ell^{(k)} = \mu_\ell^{(k-1)} - \frac{\gamma}{n}\hat{\boldsymbol{\delta}}_{\mu_\ell,i_k}^{(k)} \quad \text{and} \quad \sigma^{(k)} = \sigma^{(k-1)} - \frac{\gamma}{n}\hat{\boldsymbol{\delta}}_{\sigma,i_k}^{(k)},$$

where the learning rate $\gamma = 10^{-3}$.

**Monte Carlo Momentum update**: Set

$$\mu_\ell^{(k)} = \mu_\ell^{(k-1)} + \hat{\boldsymbol{v}}_{\mu_\ell}^{(k)} \quad \text{and} \quad \sigma^{(k)} = \sigma^{(k-1)} + \hat{\boldsymbol{v}}_\sigma^{(k)},$$

where

$$\hat{\boldsymbol{v}}_{\mu_\ell,i}^{(k)} = \alpha\hat{\boldsymbol{v}}_{\mu_\ell}^{(k-1)} - \frac{\gamma}{n}\hat{\boldsymbol{\delta}}_{\mu_\ell,i_k}^{(k)} \quad \text{and} \quad \hat{\boldsymbol{v}}_\sigma^{(k)} = \alpha\hat{\boldsymbol{v}}_\sigma^{(k-1)} - \frac{\gamma}{n}\hat{\boldsymbol{\delta}}_{\sigma,i_k}^{(k)},$$

where $\alpha$ and $\gamma$, respectively the momentum and the learning rates, are set to $10^{-3}$.

**Monte Carlo ADAM update**: Set

$$\mu_\ell^{(k)} = \mu_\ell^{(k-1)} - \frac{\gamma}{n}\hat{\boldsymbol{m}}_{\mu_\ell}^{(k)}/(\sqrt{\hat{\boldsymbol{m}}_{\mu_\ell}^{(k)}} + \epsilon) \quad \text{and} \quad \sigma^{(k)} = \sigma^{(k-1)} - \frac{\gamma}{n}\hat{\boldsymbol{m}}_\sigma^{(k)}/(\sqrt{\hat{\boldsymbol{m}}_\sigma^{(k)}} + \epsilon),$$

where

$$\hat{\boldsymbol{m}}_{\mu_\ell}^{(k)} = \boldsymbol{m}_{\mu_\ell}^{(k-1)}/(1-\rho_1^k) \quad \text{with} \quad \boldsymbol{m}_{\mu_\ell}^{(k)} = \rho_1\boldsymbol{m}_{\mu_\ell}^{(k-1)} + (1-\rho_1)\hat{\boldsymbol{\delta}}_{\mu_\ell,i_k}^{(k)},$$

$$\hat{\boldsymbol{v}}_{\mu_\ell}^{(k)} = \boldsymbol{v}_{\mu_\ell}^{(k-1)}/(1-\rho_2^k) \quad \text{with} \quad \boldsymbol{v}_{\mu_\ell}^{(k)} = \rho_2\boldsymbol{v}_{\mu_\ell}^{(k-1)} + (1-\rho_1)\big(\hat{\boldsymbol{\delta}}_{\sigma,i_k}^{(k)}\big)^2$$

and

$$\hat{\boldsymbol{m}}_\sigma^{(k)} = \boldsymbol{m}_\sigma^{(k-1)}/(1-\rho_1^k) \quad \text{with} \quad \boldsymbol{m}_\sigma^{(k)} = \rho_1\boldsymbol{m}_\sigma^{(k-1)} + (1-\rho_1)\hat{\boldsymbol{\delta}}_{\sigma,i_k}^{(k)},$$

$$\hat{\boldsymbol{v}}_\sigma^{(k)} = \boldsymbol{v}_\sigma^{(k-1)}/(1-\rho_2^k) \quad \text{with} \quad \boldsymbol{v}_\sigma^{(k)} = \rho_2\boldsymbol{v}_\sigma^{(k-1)} + (1-\rho_1)\big(\hat{\boldsymbol{\delta}}_{\sigma,i_k}^{(k)}\big)^2.$$

The hyperparameters are set as follows: $\gamma = 10^{-3}, \rho_1 = 0.9, \rho_2 = 0.999, \epsilon = 10^{-8}$.

### C.3 WALL CLOCK TIME

We provide Table 4, the running time for each method, plotted in Figure 2, used to train a Bayesian variant of LeNet-5 on MNIST. The incremental method as MISSO and MC-SAG displays a similar wall clock time, despite being a bit worse given (a) the initialization that requires to compute a vector of $n$ gradients kept in memory and updated through the iterations and (b) the average operation for each parameters update to compute the aggregated drift term, see (44).

Table 4: Bayesian Deep Neural Network: running time in seconds for 100 epochs.

|  | MC-Adam | MC-Momentum | BBB | MC-SAG | MISSO |
|---|---|---|---|---|---|
| LeNet-5 on MNIST | 12889 | 12816 | 12690 | 13822 | 13367 |

We plot Figure 4, the learning curves for the MNIST example against the time elapsed (in seconds).

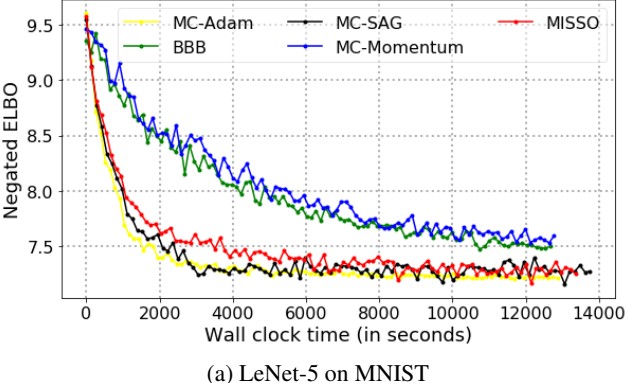

(a) LeNet-5 on MNIST

Figure 4: Negated ELBO versus wall clock time for fitting a Bayesian LeNet-5 on MNIST. Plotted on the average of the 5 repetitions.

