# OpenReview forum: "MISSO: Minimization by Incremental Stochastic Surrogate Optimization for Large Scale Nonconvex and Nonsmooth Problems"
_ICLR.cc/2021/Conference — Reject_

### Official Review · AnonReviewer4 · 2020-10-16
**Marginally above the acceptance threshold**

**Rating:** 6
**Confidence:** 3

**Review:**

1. Summarize what the paper claims to do/contribute. Be positive and generous.

In this paper, the authors consider solving the optimization of the summation of a finite number of component functions. The proposed algorithm is based on a previous work called Minimization by Incremental Surrogate Optimization (MISO). The MISO is a majorization minimization algorithm, which shares a similar update style of the SAG method. However, different from SAG, whose convergence is not available for nonconvex optimization, and is even very tricky in convex case, MISO enjoys a global convergence guarantee due to its majorization property. Based on this existing method, for the problems whose majorization surrogate is very hard to construct, e.g. variational inference of latent variable models, the authors of this paper propose a sample average approximation of the exact majorization surrogate function. The convergence of the proposed algorithm is also provided in this paper.

2. Clearly state your decision (accept or reject) with one or two key reasons for this choice.
This paper is marginally below the acceptance threshold.

3. Provide supporting arguments for the reasons for the decision.

(i). (Weakness) For the hard cases where each component is an expectation itself, the strategy applied here is to do a simple sample average approximation. This requires the sample size of in each iteration (M_k) to satisfy the condition that \sum_k M_k^{-1/2}<\infty. That is, in the $k$-th iteration, the sample size will be at least k^2. According to Theorem 1, the number of iteration should be K\geq nL/\epsilon^2. Consequently, the total sample complexity of this method seems to be \sum_{i=1}^{K} k^2 ~ n^3L^3\epsilon^{-6}. The $n^3L^3$ dependence seems very bad. However, let us do a simple estimation of a naive method: 1. In each step compute the \epsilon-accurate estimation of the gradient for each component, this needs O(n \epsilon^{-2}) samples per iteration. Then if the function is L-smooth (this paper can handle nonsmooth cases) then the total iterations will be O(L\epsilon^{-2}). Then the total sample complexity seems only O(nL\epsilon^{-4}). This might need some clarification.

(ii). (Strength) This paper provides a non-asymptotic rate of convergence for the MISSO algorithm, which implies a non-asymptotic rate for the MISO method, whose non-asymptotic rate is not known before, which should be appreciated. Moreover, the numerical experiment in this paper is well presented.

4. Provide additional feedback with the aim to improve the paper. Make it clear that these points are here to help, and not necessarily part of your decision assessment.

(i). The MISSO (and MISO) share a similar updating style with SAG, it will be better if the authors could add some discussion on their relation and difference. Or, if such discussion exists in other literature, add a reference to that.

(ii). After the Theorem 2. It may make sense to give the sample complexity of the result. Namely, to get the optimality measure \leq \epsilon, how many sampled are needed. Specifically, by the reviewer’s rough estimation, the dependence on n and L is O(n^3L^3), see my argument before, this dependence is not reasonable. My question is that can the authors carefully balance the parameters and derive a more reasonable sample complexity? If the O(n) and O(L) dependence can be achieved, the reviewer is willing to change to a higher score.

---

> ### Author Response · Authors · 2020-11-13
> **Response to AnonReviewer4**
>
> We thank the reviewer for his/her critical, yet very helpful comments. Below, we try our best to address your concerns on the significance of the results of our paper.
>
> Firstly, we agree with the reviewer that in the \textbf{special case of quadratic surrogates}, the MISSO update yields to a gradient update very similar to the SAG [Le Roux, Schmidt, Bach, 2012]. Yet, the authors would like to draw attention on a subtle difference here. In the MISSO method, since the minimization occurs on the aggregate sum of the stochastic surrogates, a simple derivation of the minimization of quadratic functions gives this term as being equal to the mean of the past $n$ iterates (i.e. $1/n \sum_{i=1}^n \theta^{\tau_i^k}$). Whereas in SAG, as any variance reduction technique, the contribution is in the drift term (constructed through incremental update) leaving the first term unchanged vis-a-vis SGD as equal to the last iterate (i.e., $\theta^{k-1}$).
> Of course when the user designed stochastic functions are no longer quadratic, the parallel with any stochastic gradient methods is no longer available, thereby making our framework more general.
>
> Secondly, we are glad that the reviewer has brought up the issue on sample complexity, which is an important metric that we have missed in the first version. We give below a clarification on the said issue. As the reviewer suggested, we limit ourselves to smooth optimization and study the number of samples needed to attain a stationary solution with $|| \nabla L( \theta) ||^2 \leq \epsilon$. In this setting, the complexity for the naive SGD method described by the reviewer has a complexity of $O( nL / \epsilon^4)$.
>
> On the other hand, for MISSO, we note that the stationarity metric in (16) satisfies $|g_{-}(\theta)| = || \nabla L(\theta) ||$. As such, to make a fair comparison with the above, it is important to consider $|g_{-}(\theta)|^2$ and compare the number of iterations needed to attain a $\epsilon$ stationary solution should be $O( \Delta / \epsilon)$, where $\Delta$ is defined in Theorem 1.
>
> Now, in order to keep $\Delta \asymp nL$, we need to set $M_k = k^2 / n^2$, therefore this yields the sample complexity of $\sum_{k=1}^{K_{max}} M_k = (1/n^2) \sum_{k=1}^{ O(nL/\epsilon) } k^2 = O( n L^3 / \epsilon^3 )$.
> In conclusion, we found that the *sample complexity* for MISSO is also lower in terms of the dependence on $\epsilon$, though it comes at a price of an additional $L^2$ term.
>
> In summary, we have:
>
> *Iteration Complexity:* According to Theorem 1, MISSO requires $K = {\cal O} (n L/\epsilon)$ iterations to ensure $|g_-( \theta^{K} )|^2 < \epsilon$ ($\epsilon$-stationarity).
> Whereas, for the naive algorithm proposed by the reviewer, with batch setting it requires $K= L/\epsilon^{2}$ iterations to get $\epsilon$-stationarity.
>
> *Sample Complexity:* For the naive method, the sample complexity of $O( nL/\epsilon^4 )$ holds.
> Yet, for MISSO, if we set $M_k = k^2/n^2$ such that $\Delta$ is of order $\mathcal{O}(nL)$, the sample complexity becomes $\sum_{k=0}^{nL/\epsilon} k^2/n^2 = (1/n^2)*(nL/\epsilon)^3 = nL^3 / \epsilon^3$. In comparison with the proposed method ($nL/\epsilon^4$), we sacrifice $L^2$ to win an order of $\epsilon$.
> We will include the above calculations in the revised paper. Nevertheless, the authors are happy that the reviewer has raised the above concerns, which led us to further examine the benefits of the MISSO method.

---

### Official Review · AnonReviewer2 · 2020-10-28
**A stochastic optimization method with surrogate functions from Monte Carlo samples is developed**

**Rating:** 5
**Confidence:** 1

**Review:**

This paper proposed MISSO, which is an extension of MISO to handle surrogate functions that are expressed as an expectation. MISSO just used the Monte Carlo samples from the distribution to construct objectives to minimize.

It seems to me that MISSO is just a straigforward extension of MISO, also the empirical results seems to suggest the proposed MISSO has no advantage over Monte Carlo variants of other optimizers, such as MC-SAG, MC-ADAM, thus it is not clear to me what is the significant aspect of this work.

---

> ### Author Response · Authors · 2020-11-13
> **Response to AnonReviewer2**
>
> We thank the reviewer for his/her comments. Below, we try our best to address your concerns on the originality of our paper.
> We want to stress that the paper's significance lies on the generality of our incremental optimization framework, which tackles a constrained, non-convex and non-smooth optimization problem.
> The main contribution of this paper is to propose and analyze a unifying framework for a large class of optimization algorithms which includes many well-known but not so well-studied algorithms.
>
> The major idea here is to relax the class of surrogate functions used in MISO [Mairal, 2015] and to allow for intractable surrogate that can only be evaluated by Monte-Carlo approximations.
> We provide a general algorithm and global convergence rate analysis under mild assumptions on the model and show that two examples, MLE for latent data models and Variational Inference, are its special instances. Importantly, our convergence analysis applies to *both* applications for which analysis are lacking in the current literature.
> The major proof idea here is to relax the class of surrogate functions used in MISO [Mairal, 2015] and to allow for intractable surrogate that can only be evaluated by Monte-Carlo approximations. Working at the crossroads of Optimization and Sampling constitutes what we believe to be the novelty and the technicality of our results.

---

### Official Review · AnonReviewer3 · 2020-10-28
**A good paper**

**Rating:** 7
**Confidence:** 3

**Review:**

This paper propose a doubly stochastic MM method based on Monte Carlo approximation of these stochastic surrogates for solving nonconvex and nonsmooth optimization problems. The proposed method iteratively selects a batch of functions
at random at each iteration and minimize the accumulated surrogate functions (which are expressed as an expectation). They establish asymptotic and non-asymptotic convergence of the proposed algorithm. They apply their method for inference of logistic regression model and for variational inference of Bayesian CNN on the real-word data sets.

Weak Points.
W1. The authors do not discuss the connections with state-of-the-art second-order optimization algorithms such as K-FAC.
W2. The proposed algorithm still falls into the framework of MM algorithm and a simple convex quadratic surrogate function is considered. The convergence rate of the algorithm is expected.

Strong Points.
S1. The proposed method can be viewed as a combination of MM and stochastic gradient method with variance reduction, which explains its good performance.
S2. The paper contains sufficient details of the choice of the surrogate function and all the compared methods in the experiments.
S3. The authors establish asymptotic and non-asymptotic convergence of the proposed algorithm. I found the technical quality is very high.
S4. Extensive experiments on binary logistic regression with missing values and Bayesian CNN have been conducted.

---

> ### Author Response · Authors · 2020-11-13
> **Response to AnonReviewer3**
>
> We thank the reviewer for his/her interest in our paper. Below we address your concerns about our weaknesses.
>
> Firstly, we believe that the research direction on second order surrogates is an interesting one. Particularly, papers such as K-FAC ("Optimizing Neural Networks with Kronecker-factored Approximate Curvature" by Martens and Grosse) would be an interesting comparison, as it involves using an approximation of the Hessian matrix. We believe that the MISSO method can include ideas similar to K-FAC as a special case. In particular, it can be done by considering a *scaled* quadratic surrogate function, e.g., for the VI example in (11), we may replace the last term by $|| \bar{\theta} - \theta ||_{H}^2$, where $H$ is an approximate Hessian. In addition, we also refer to the recent work on "IQN: An incremental quasi-Newton method with local superlinear convergence rate." by Mokhtari, Eisen, and Ribeiro, in SIOPT, 2018, where a BFGS like method using memorized quantities to reduce the variance of stochastic approximations is applied to the problem of stochastic optimization leveraging quasi-Newton functions. Their work is on (a) convex and strongly convex functions and (b) deterministic surrogates.
>
> Secondly, we notice that the MM algorithm is in fact very general where the only requirement on the surrogate function is that it satisfies H1,H2 (on page 2), and all our analysis afterwards will follow. As discussed in (Mairal, 2015), we believe that the MM is sufficiently general to be considered relevant to the ML community. Due to the limitation of space (in the main paper), we have only explicitly stated the quadratic surrogate function used for the VI example. However, in Example 1: MLE example via stochastic EM, a different surrogate function is actually used under the hood, and the reviewer is referred to section B.2 for a detailed discussion. Finally, we notice that convergence in expectation is commonly established in the optimization literature and our notion of convergence is standard. Furthermore, we notice that the result can be easily converted to that of high probability convergence using the classical Markov inequality.

---

### Official Review · AnonReviewer1 · 2020-10-29
**Interesting theory, though the practical relevance is unclear**

**Rating:** 6
**Confidence:** 4

**Review:**

This manuscript contributes a stochastic optimization method for finite sums where the loss function is itself an intractable expectation. It builds upon stochastic majorization-minimizations methods, in particular MISO, that it extends to use Monte-Carlo approximation of the loss.

I am happy to see some attention put to the majorization-minimizations methods, which have many interesting benefits. The paper contributes nice theoretical results, in particular non-asymptotic results. However, I believe that these theoretical results are not enough to situate the contribution with regards to the wider landscape of optimization methods for machine learning.

In this respect, the empirical study is crucial, however it is not completely convincing. Expressing figures 1 and 2 as a function of the number of epoch, rather than as an estimate of runtime is not meaningful: it discards the cost of running the inner loop, which varies from one approach to another. It would leed to believe that MISSO50 is the best option, which is probably not the case.

Also, MC-ADAM seems to outperform MISSO for variational inference

With regards to the broader contribution, it is very appreciable to have a wider theory of stochastic optimization with MM methods. It would have been good, however, to have a discussion of the link of the contributed method to the follow up work by Mairal and colleagues, Stochastic Approximate MM (Mensch et al 2017).


**Additional comments after the discussion**

The authors have thoroughly replied to all the comments from the various reviewers.

After reading all the discussions (other reviews as well as replies from the authors), it appears to me that the practical relevance of this contribution is not completely clear. The computational cost of each iteration is large. The benchmarks do not show clear improvements in computational.

---

> ### Author Response · Authors · 2020-11-13
> **Response to AnonReviewer1**
>
> We are grateful for the reviewer who has found the studied subject on MM methods interesting. We too believe that the MM method is a versatile framework for tackling modern ML problems and deserve more attention in this community.
>
> In our experiments, we found that the tested methods involve a similar number of gradient computations per iteration (since reported every epoch), as such the wall clock time per iteration are comparable. To further support this comment, in the revised version, we provide in the appendix additional comparison of the convergence against the running time.
> Also, MC-ADAM seems to outperform MISSO for variational inference
> We agree with the reviewer that MC-ADAM seems to outperform MISSO on the variational inference example. Indeed, we must acknowledge that while our MISSO scheme does not beat the SOTA (such as MC-ADAM) in every example. However, we should emphasize that the goal of this paper is to propose a simple yet general incremental optimization framework which encompasses several existing algorithms for large-scale data. In particular, the same framework also applies to other learning problems such as MLE with stochastic EM (Example 1).
>
> -- " link of the contributed method to the follow up work by Mairal and colleagues, Stochastic Approximate MM (Mensch et al 2017)"
>
> We believe the reviewer makes a reference to "Stochastic Subsampling for Factorizing Huge Matrices" by Mensch et. al. (https://arxiv.org/pdf/1701.05363.pdf). In this paper, the authors focus on the problem of matrix factorization for the purpose of dictionary learning. In the particular and challenging case of sparsity and high dimensional matrices (typical in fMRI data), the authors propose a stochastic MM scheme. The level of stochasticity occurs in the index sampling step (sampling subset of dimensions), see first step in Algorithm 3 in their paper, then compute the parameters of the surrogate function leveraging a deterministic (no added stochasticity in this step) Robbins-Monro type of update.
> Rather, in our work, two levels of stochasticity are at stake. The first one is similar as theirs, i.e. the sampling of individual indices, and the second one deals with the Monte Carlo approximation of the intractable surrogate functions (written in our illustrative examples as expectations). The theoretical and practical study of a doubly stochastic as MISSO constitute the main contributions of our paper compared to the mentioned reference, while sharing similar assumptions on the model such as smoothness and existence of directional derivative (see their assumptions (D) and (E)).
> In the revised version, we will include a discussion on the above mentioned works.

---

### Official Review · AnonReviewer5 · 2020-11-09
**Reviewer 5's Report**

**Rating:** 3
**Confidence:** 5

**Review:**

This paper develops a stochastic MM-type algorithm to minimize a finite sum. Essentially, the stochastic method draws one sample at each iteration, and find a majorization surrogate for the corresponding loss, and find the minimizer for the updated total loss.

Overall, I don't find the paper well-developed and doesn't meet the bar of a top conference like ICLR for the following major concerns:

1. The major flaw is that in each iteration, the algorithm requires us to find the minimizer of the updated total loss (Step 8 of algorithm 2). This step is computationally as expensive as the update step in a batched MM algorithm. For a stochastic-type algorithm, I would expect the update only finds the minimizer of the stochastically picked individual surrogate function.

2. By minimizing a stochastically picked individual surrogate function, the convergence follows by existing literature on stochastic proximal gradient method, there Theorem 2 follows without much difficulty.

3. The convergence rate of the proposed method is not derived, which shouldn't be too difficult to derive.

---

> ### Author Response · Authors · 2020-11-13
> **Response to AnonReviewer5**
>
> We thank the reviewer for his/her critical comments. However, we believe that our paper does contain a number of important contributions, as we try to explain below
>
> As rightfully mentioned, the M-step (line 8 of Alg. 2) can be as costly as the batch MM method *in the worst case*.
> However, in many cases, the MISSO algorithm can take advantage of the (stochastic) surrogate function's structure such that the M-step can be performed in low complexity. Notice that having an easy-to-optimize surrogate function is often a major advantage with MM methods.
> Let us first consider Example 2: the VI example (in p.4), the surrogate functions are quadratic approximations of the likelihood functions. Here, the M-step updates can be derived in closed form involving a running average of the previously drawn parameters and samples (e.g., it has a similar form as popular methods such as SAG, SAGA). Thus, the complexity of line 8 is similar to any stochastic method, i.e., independent of $n$
>
> Likewise for Example 1: MLE with stochastic EM (in p.3), where, since the complete log likelihood belongs to the curved exponential family, the opaque M-step (line 8) is expressed w.r.t the sufficient statistics, see Section B.3 of the supplementary material. Hence, the M-step actually leverages the incremental characteristic of MISSO since the stochastic sufficient statistics used in the M-step are also incrementally updated.
> The advantage of our incremental method, MISSO, lies in the use of line 7 where only a mini batch of stochastic surrogates is updated, with the majority of surrogates being unchanged.
> For your suggestion of having updates that "only find the minimizer of stochastically picked individual surrogate function", we believe that this will lead to an algorithm similar to plain SGD instead of the incremental stochastic methods like MISSO. To see this, we can consider employing a stochastically picked (say the $i_k$th function) quadratic, deterministic surrogate expanded around $\theta^k$, then we update $\theta^{k+1}$ as a convex combination of $\theta^k$ and the minimizer to the surrogate. It can be shown that
> $\theta^{k+1} = \theta^k - \gamma_k (1/L) \nabla L_{i_k}( \theta^k )$
> where $\gamma_k \in [0,1]$ is the convex combination weight and we have used that the surrogate's minimizer is $\theta^k - (1/L) \nabla L_{i_k}( \theta^k )$.
> In this way, the algorithm cannot take advantage of the finite sum structure of the optimization problem and the convergence rate can be worse
>
> We emphasize that there are two sources of stochasticity in our MISSO framework (and settings in general). Not only is the individual surrogate function stochastically picked (as stated by the reviewer), but the latter is also approximated by Monte Carlo sampling. This double level of stochasticity makes most of the existing theoretical convergence proofs inapplicable for our work.
> We notice that in addition to Th.2 which establishes the asymptotic convergence of MISSO, our Th.1 also provides the non-asymptotic convergence rate for MISSO. We must stress that compared to Th.1, the existing results on stochastic proximal gradient algorithms are not sufficient to provide such almost sure convergence guarantee. Indeed, we shall recall to the reviewers that our MISSO framework is strictly more general than stochastic gradient method as it also includes, among others, EM-like algorithms that can not always be casted as gradient methods.
> As such, we respectfully disagree with the reviewer that our "convergence follows by existing literature on stochastic proximal gradient" due to the vast difference between MM methods and standard gradient-based method. In particular, although stochastic proximal gradient methods can be regarded as a special case of MISSO, the latter method is very different and requires different analysis tools. As a comparison, in the recent work on "Proximal-proximal-gradient method" by Ryu and Yin, Journal of Computational Mathematics, 2019, the authors proposed a general framework to unify proximal gradients algorithms and cast them as MM methods (for deterministic and convex objectives), yet the method uses stepsizes of order 1/L rather than n/L.
>
> We have indeed derived the non-asymptotic convergence rate for our MISSO method in Th.1.
> It gives a global rate on (1) the gradient of the gap between the surrogate and the objective function and (2) the stationary condition (eq. (14))
>
> Th.1 explicitly shows the convergence of MISSO "at a sublinear rate $\mathbb{E}[ g_-^{(K)} ] \leq {\cal O}( \sqrt{1 / K_{\sf max}} )$". Hence, MISSO requires ${\cal O} (n L/\epsilon)$ iterations to ensure $||g_-( \theta^{K} )|| < \epsilon$ (as a definition of $\epsilon$-stationarity for *constrained* optimization). Notice that the obtained rate of convergence is comparable to existing algorithms on non-smooth optimization that rely on processing a full batch of data at each iteration. In the revision, we also discuss the sampling complexity of MISSO

---

### Decision · Program_Chairs · 2021-01-07
**Final Decision**

**Decision:**

Reject

**Comment:**

This paper extends the Majorization Minimization principle, particularly the MISO method, to problems where the surrogate of randomly selected batch functions are intractable, such as in formulations rising from Variational Inference. There is a large gap between the reviewers' evaluations even after the author rebuttal and discussions. While the strength of the proposed method looks to be its generality, the main criticism from the reviews are the limited applicability and less convincing arguments and empirical evidences against alternatives such as Monte Carlo versions of popular adaptive stochastic optimization methods. Weighing these considerations and considering strengths of other submissions on similar topics, I have to recommend rejection of the paper at the current form.